# Integrative analysis of yeast colony growth
Tünde Gaizer [1,3], János Juhász[1,2,3], Bíborka Pillér[1], Helga Szakadáti[1], Csaba I. Pongor[1] & Attila Csikász-Nagy [1] ✉

Yeast colonies are routinely grown on agar plates in everyday experimental settings to understand basic molecular processes, produce novel drugs, improve health, and so on. Standardized conditions ensure these colonies grow in a reproducible fashion, while in nature microbes are under a constantly changing environment. Here we combine the power of computational simulations and laboratory experiments to investigate the impact of non-standard environmental factors on colony growth. We present the developement and parameterization of a quantitative agent-based model for yeast colony growth to reproduce measurements on colony size and cell number in a colony at non-standard environmental conditions. Specifically, we establish experimental conditions that mimic the effects of humidity changes and nutrient gradients. Our results show how colony growth is affected by moisture changes, nutrient availability, and initial colony inoculation conditions. We show that initial colony spread, not initial cell number have higher impact on the final size and cell number of colonies. Parameters of the model were identified by fitting these experiments and the fitted model gives guidance to establish conditions which enable unlimited growth of yeast colonies.

The budding yeast *Saccharomyces cerevisiae* is one of the most commonly used microorganisms in the food and biotechnology industry and also serves as a model organism to understand basic biological processes. *S. cerevisiae* has contributed to the understanding of some of the most fundamental molecular processes, from cell cycle regulation to autophagy and so on[1–3]. Growing yeast or bacterial colonies on semi-solid surfaces is an everyday task in most laboratories. The first detailed investigation of colony growth dynamics dates back to the 1920's, when the linear growth model of colony diameter was established[4]. In the 1960s Pirt extended this model by an initial 12 h exponential growth period. Later Gray and Kirwan[5] showed that diffusion can limit colony growth and the diameter of the dish can influence the maximal size of a colony. They have also realized that colonies grown in proximity to each other can affect their growth kinetics. In the 90's colony growth was characterized as a biphasic process by Meunier and Choder[6], demonstrating that an initial rapid growth phase is followed by a sharp transition to a slower growth phase due to the cells in the center going into a stationary phase and growth being driven dominantly by peripheral cells. Further aging of the colony leads to a scarcity of nutrients and oxygen inside, resulting in differentiation within the colony[7]. These days, growing yeast colonies and using colony size as an indicator of fitness is a daily routine in many research laboratories[8,9]. Still, the underlying cellular processes which affect colony growth are far less understood. Most of the above-mentioned quantitative studies focus on the measurements of colony diameters and only a little is known about the processes which happen between the cells inside yeast colonies[10–14] Ammonia was identified as a

communication factor between yeast colonies[15,16], and emergence of metabolic and transcriptional heterogeneity inside yeast colonies were also identified[17,18]. More is known about the colony growth regulation of bacterial colonies. Metabolic and electrical signaling inside and between colonies has been identified[11–14], but similar depth studies have not been performed on yeast colonies. We know about interactions between and inside colonies, and we know that yeasts evolved in a constantly changing environment, while in laboratory experiments, we minimize the variations in conditions (media, temperature, humidity, seeding conditions, light) for standardization and reproducibility. With these restrictions we cannot fully reveal how yeast colonies are formed, grow and interact. Mathematical models have helped the interpretation of many of the above listed observations on microbial colony growth[13,19] and others also focused on integrating knowledge to capture colony growth dynamics[20–22] and colony structure emergence[23–26], even incorporating physical forces[27–29]. Some models consider individual microbes as agents, which interact with their environment and neighboring cells[30]. Such agent-based (or individual-based) modeling techniques were also used to understand how bacterial colonies grow[21,31] and spatial patterns in their form might emerge[32] or to investigate microbial communities[33,34]. While there is a high level of similarity between the structure of bacterial and yeast colonies, there are some considerable differences in how these are formed. Motility of bacterial cells is missing in yeasts, and their cell division characteristics and periods are also dissimilar. These differences need to be taken into consideration before applying the above bacterial models for yeast colony formation.

[1]Pázmány Péter Catholic University, Faculty of Information Technology and Bionics, Budapest, Hungary. [2]Semmelweis University, Institute of Medical Microbiology, Budapest, Hungary. [3]These authors contributed equally: Tünde Gaizer, János Juhász. ✉e-mail: csikasz-nagy.attila@itk.ppke.hu

Despite of these efforts, our current knowledge still lacks quantitative details on how growth of colonies depend on diffusibility of media in the agar and how initial layouts of cells can influence colony growth. Here we combine quantitative colony measurements in various conditions with agent-based modeling to reveal the dynamical features and limiting factors of colony growth. Specifically, we parameterize an agent-based model to fit measurements of yeast colony growth dynamics at various initial densities, layouts and agar media conditions. Then we use the model to predict how diffusion and proximity of other colonies can limit colony growth. The model helped us to come up with conditions to grow unconditionally large colonies and show that initial spread of cells have a greater influence of colony size than initial cell number has.

## Results

### An agent-based model of yeast colony growth

To be able to capture the dynamics and limiting factors of yeast colony growth we turned to agent-based modeling[33,35], where each cell is considered as an agent interacting with its environment and following a basic growth cycle. The model considers yeast cell groups as agents with their main goals to grow and divide. The metabolism of the cells is simplified to an energetic need provided by one nutrient present in the environment.

Each cellular agent has a life cycle evaluated in every simulation step (Fig. 1a). This life cycle has the following stages: The agent takes up a predefined amount of nutrients from the layer grid cell it is situated on and adds it to its energy level. This energy level is used for covering resources needed for metabolism and growth. The metabolic energy is subtracted from the cell in each simulation step. If the agent's energy level falls below a certain threshold, the agent dies, becomes inactive, and gets removed from the simulation. Its remaining energy returns to the layer as a nutrient. If it reaches a predefined energy level (division threshold) the cell divides into two cells, the daughter cell gets the initial cellular energy level (and starts its own life cycle), and the rest remains in the mother cell. Cell division happens

in two dimensions: the daughter cell gets a random position at a certain distance from the mother cell. The daughter cell has the same properties as the mother cell except for its position and energy level. This asymmetric cell division reflects the typical behavior of budding yeast[36]. If the energy level of the agent is between a given threshold (G0 threshold) and the death threshold it enters a passive, solitary, G0 state from the previous, active state. An agent in this G0 state[37,38] cannot divide and consumes less nutrient/energy than in its active, dividing state. It is beneficial if the agent is in a nutrient-deficient environment because it can survive longer with baseline metabolism (and can reactivate itself if the nutrient level increases again) (Fig. 1a).

The agar plate is simulated as a two-layer square grid plane (of nutrients) in the model, on which the agents (yeast cells) are positioned. The upper layer feeds the cells in the growing colony and the lower layer feeds the upper one and ensures that nutrients can diffuse even to inner regions of the colonies where many cells are actively growing. This setup serves as a computationally efficient rough approximation for the three-dimensional nature of the media. The agents are immobile and point-like represented by their coordinates on the plate. It indicates that they do not have an explicit area. Multiple agents can be on a grid cell, but the agents interact (take up nutrients) directly only with their grid cell. Agent density develops according to local nutrient supply and availability as an emergent property. The nutrient concentration is tracked in each grid cell of both layers and this concentration changes in every simulation step by nutrient uptake (via the cellular agents), by diffusion inside the layers, and by nutrient transport between the layers. In this way, neighboring cells, which are present in a single or nearby grid, are competing for the locally present nutrients which can be depleted, but slowly recharged from the bottom layer. Parameters of the plate (the environment), initial cell layout, and yeast strains can be easily configured through an input parameter file (See Code availability). This enables placing multiple colonies on a simulation plate (currently up to 24 but can be upscaled for any layout). A detailed description of the model

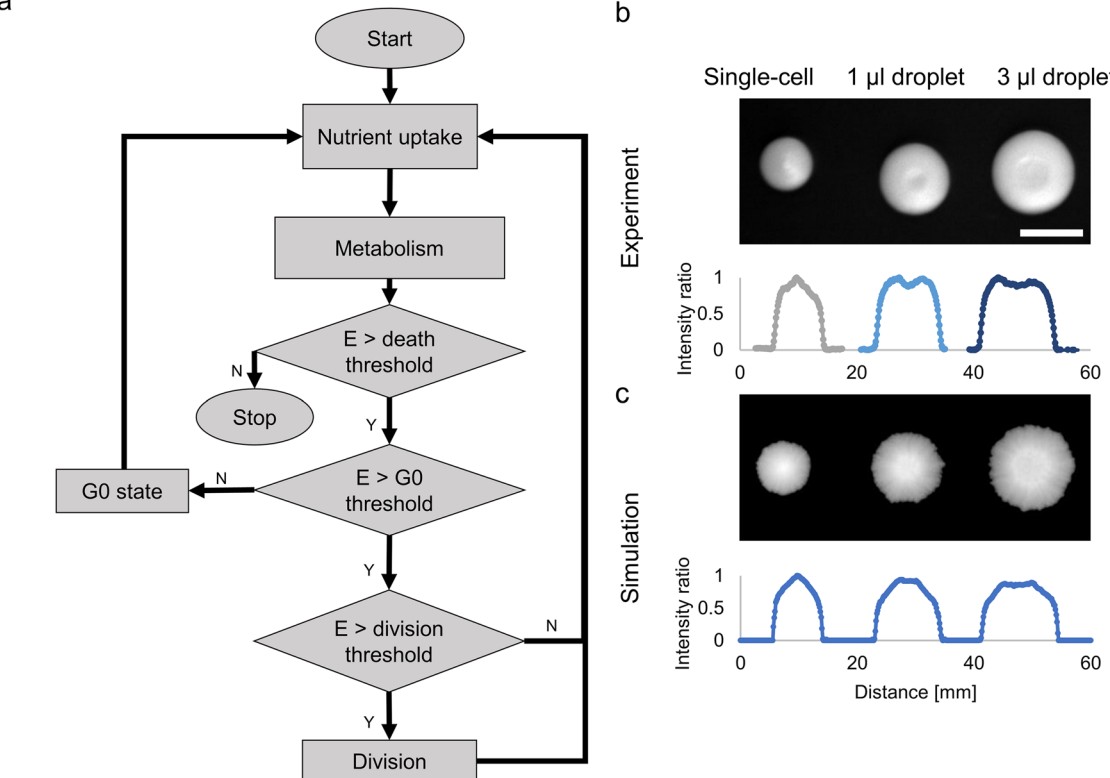

**Fig. 1 | An agent-based model of yeast colony growth. a** Core life cycle algorithm of the agent-based model for each simulated yeast cell. **b** Colonies grown from a single-cell and from 1 and 3 µl droplets at the age of 8 days and their cross-sectional intensity profiles. Scale bar represents 1 cm. (Representative images from $n = 4$). **c** Illustrative simulations of the above experiments and their cross-sectional intensity profiles. Conversion: 55.6 simulated area units per mm².

parameters, their values and the primary outputs can be found in the Methods and Supplementary Table 2 and 3.

The qualitative capabilities of the model were tested and validated with experiments in multiple aspects (Colony shape, size, and behavior of multiple colonies next to each other). First, we tested the model on fitting colony sizes and shapes formed after various inoculations. Single-cell colonies (simulated colonies initialized with one agent) and colonies inoculated from a droplet show distinct colony shapes. When considering local cell density as a third dimension, the simulations represent well the conical shape of a single cell colony[39] and the volcano-like shapes experienced when inoculating with liquid droplets[40]. The difference in the crater size depending on the inoculation droplet size is also apparent (Fig. 1b). Yeasts, especially wild isolate strains are capable of forming biofilms and more complex 3-dimensional structures that depend on local/physical forces[25,41] or cell differentiation. Currently, these forces are not incorporated into the model.

### Growth dynamics parametrization

To determine the agent/strain-specific and environment-specific parameters of the model, first, we investigated the limitation of nutrients in a plate when multiple colonies are inoculated on it. Multiple colonies (1, 2, 3, 4, and 6) were inoculated on a single Petri dish as shown in Fig. 2a. Colonies were grown for 20 days for both qualitative and quantitative analysis. Growth curves (Fig. 2c) show that the time of deceleration of the growth rate and entering in the stationary phase depends inversely on the number of colonies. One can observe that colony shapes change from circular to a less and less circular shape due to the proximity of neighboring colonies and the edge of the plate. This feature was also present in our simulations (Fig. 2b, Supplementary Fig. 1). Besides fitting the area and considering the shape of the colony, this experimental setup provided the ground for parametrizing the model to represent the distance in which neighboring colonies start to inhibit each other by competing for nutrients. The area over time was used to determine the following parameters of the model: cell division distances, nutrient diffusion rates, initial nutrient levels, and initial colony sizes.

Cell division distance parameter defines distance between the daughter cell and its mother cell after division. The size of the colony and the density of the agents inside the simulated colonies can be controlled by modifying this parameter. Its value is expressed in grid cells, so it is dependent on the spatial resolution of the model and the size of the simulated plate.

The initial colony size parameter defines the radius of the circular region populated with cells belonging to the same strain (having the same properties) at the beginning of the simulation. It reflects the initial drop size at the surface of the agar plate and has considerable impact on the final colony size. Cell concentration, volume of the initial drop, the height from which the drop falls onto the agar and the viscosity of the agar surface all influence the drop radius in a complex way, so it was more efficient to fit this parameter instead of calculating it. Its value is also expressed in grid cells, so it is dependent on the spatial resolution of the model and the size of simulated plate. See Supplementary Table 3 for the parameter values.

Initial nutrient level parameters of the two nutrient layers define the nutrient content of each grid cell in the upper and in the lower layer at the beginning of the simulation. It is defined in abstract units as the other nutrient and energy connected terms in the model. Nutrient diffusion rates set the speed of material flow within the layers. High rates make nutrient supply fast for the cells while low rates enable the emergence of nutrient depleted zones beneath and around the communities easier and earlier. These two sets of parameters influence the speed and duration of the active growth phase of the colony development as well as the number of agents. They are connected to the nutrient uptake and other cell metabolism parameters by providing external input for the cells. Their values were different in different experiments (Supplementary Table 3) and although the tendencies were known (more or less available nutrient, more or less dense agar media) their exact values were not determined instead of fitted for the colony growth data.

Descriptions and default levels of the model parameters can be found in Supplementary Table 2. A summary of the parameter values that vary between the experiments is shown in Supplementary Table 3.

### Studying initial conditions of inoculation

During early runs of the model, initial colony seeding arose as a critical condition. To test the importance of initial seeding conditions for the colony size, we have constructed an experiment in a standard rectangular plate format (OmniTray) to investigate two initial inoculation parameters: droplet size and cell number. Inoculation droplet size was varied by changing the volume of the drops while cell numbers in these drops were fixed by changing the concentrations accordingly. The other variable, the initial cell count, was varied by changing the concentration, while keeping the drop sizes. For a combined design to test these two parameters, a $4 \times 6$ grid was applied with increasing initial droplet sizes and inoculating cell counts as shown in Fig. 3a. Proportional differences in the colony size on the second day (blue circles in Fig. 3a) were present due to the spreading of the droplets according to their volume (0.5 to 5 µl). 5 µl colonies on the second day were on average 2.8x larger than the colonies starting from 0.5 µl initial droplet size, while this difference was reduced to 1.4 by the 12th day of the experiment. The aforementioned colony layout was simulated as well with

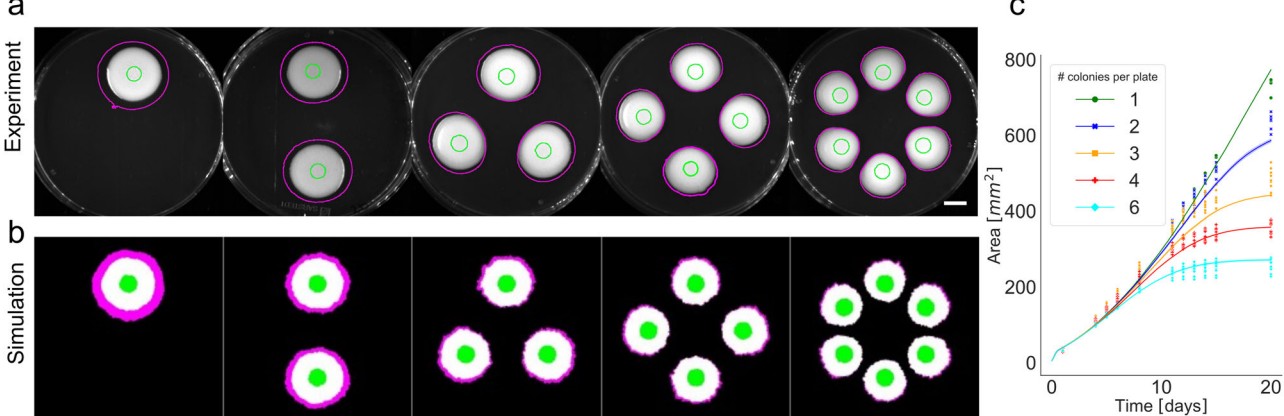

**Fig. 2 | Impact of adjacent colonies on each other's growth kinetics. a** Increasing number of colonies were inoculated on standard Petri dishes (1,2,3,4, and 6 colonies per plate) (sample sizes in order: 4,6,9,12,12). Pictures shown were taken 13 days after inoculation, green circles indicate area after 1 day and magenta circles indicate area after 20 days (Scale bar represents 1 cm). **b** Simulation of the above experiment. Green indicates the colony area after 1 day, white after 13 days, and magenta after 20 days. **c** Kinetics of colony area by colony number on the plate is shown. Experimental data is shown with markers and simulations with a solid line. Simulation results are scaled 55.6 pixels to 1 mm².

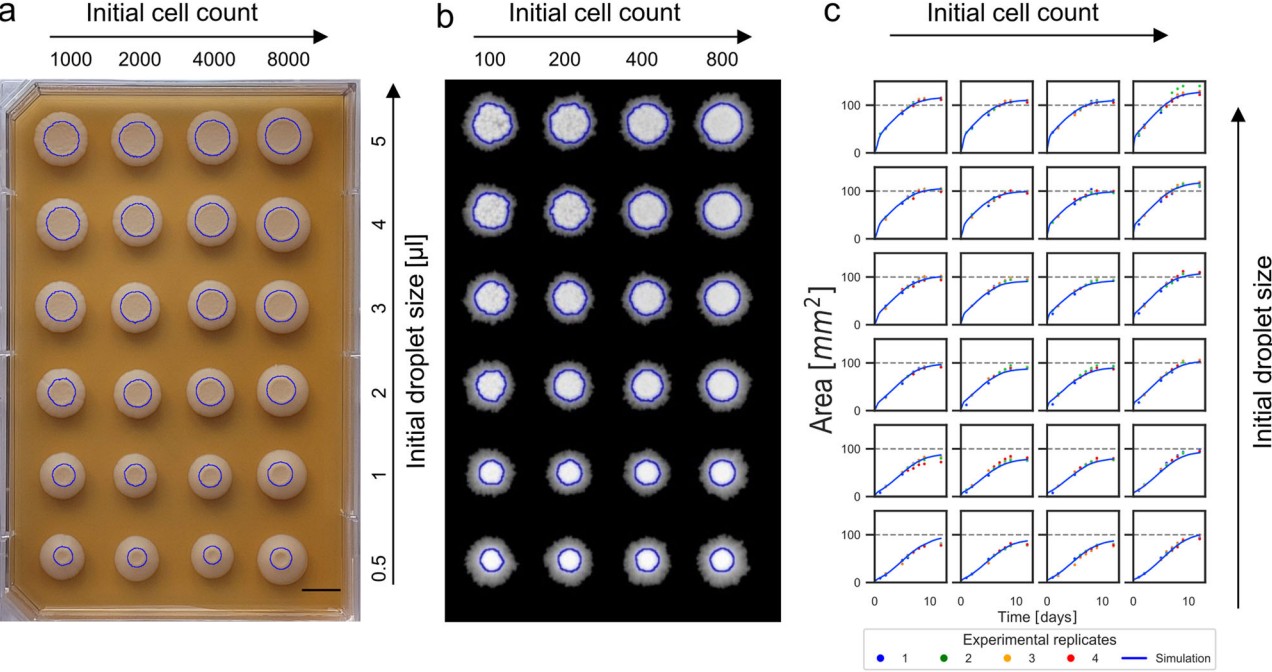

**Fig. 3 | Determining the impact of inoculation conditions. a** 24 colonies grown from various initial droplet sizes (marked μl drops) and initial cell numbers (labeled initial cell counts) after growth for 14 days. Blue circles indicate the size of the colonies on the second day which approximates the original area occupied by the inoculating droplet. (Scale bar represents 1 cm). **b** Simulation of the model with a similar layout. Blue circles indicate the area after 2 days. **c** Area growth curves of the experiment (markers of 4 replicate plates) and fitted simulations (solid lines). Dashed line at 100 mm² was inserted for easier comparison. Simulation results are scaled 55.6 pixels to 1 mm².

our computational model (Fig. 3b). To mimic the conditions of regular high-throughput OmniTray experiments, an agar layer was defined with high initial nutrient level, which is constantly consumed by cells at all spots. Diffusion values, cell division distances and initial nutrient amounts were fitted to match the simulated colony sizes to experimental data. See Supplementary Table 3 for more details on parameters.

Comparing colony sizes over a 2-week period (Fig. 3c) as well as at the final time-point (Supplementary Fig. 3) revealed that initial inoculation volume (rows) has a larger impact on colony size than cell count (columns). The far-right column with 8000 initial cells shows a prolonged growing phase, hence a larger size compared to others. This is due to the slightly asymmetrical arrangement of initial drops, enabling the right column to reach more nutrient resources. Simulations of exact copies of plate arrangements (Fig. 3b) and other arrangements of samples (Supplementary Fig. 2 and 3) confirm that this was only an edge effect.

## Understanding the effects of agar wetness on colony size and structure

Moisture is one of the environmental factors that constantly changes in nature. A lower agar density can be considered as a moist environment. It was shown that lower agar density increases colony size[42,43] and cell-cell and cell-agar surface tension was proposed to cause these differences[31,44]. To test the effects of agar wetness changes on colony growth in our model we set up an experiment with changing the environment of the agar disc upon which the colonies were growing. An agar disc was surrounded with liquid media to imitate a moist environment (wet condition). In this environment, colonies grow twice (1.95) as large in area as in a dense environment, with similar nutrient content (Fig. 4a wet condition). Besides their size, there is a notable difference in the colony structures too. The colonies in this wet condition formed a flatter, more mucous structure with edges flattening out, compared to the 'dry' condition (grown on regular 2% YPD agar) (Supplementary Fig. 4).

The experiment was constructed to imitate not just the wet and dry conditions but their alternation too, resembling changing weather conditions in nature. In this alternating condition differences in radial growth

rate, as well as a deviation in structure can be observed. The colony expands faster (Fig. 4c) in the wet period and forms a denser, better-defined structure in the dry period (Supplementary Fig. 4). The radial growth rate for the wet period in the first week was 1.09 mm/day (st. dev. 0.03) while in the dry case, it was 0.45 mm/day (st. dev. 0.01). The second phase showed similar trend, where wet conditions speed up colony expansion, with 0.97 mm/day (st. dev. 0.02) for wet conditions and 0.34 mm/day (st. dev. 0.05) for dry conditions (Supplementary Table 1). Although second-phase growth rates are lower than those in the first phase (agar is aging), the radial growth rates depend on the actual condition and not on the condition colonies have been earlier.

Total cell counts and dry masses were determined after 14 days for all 4 cases confirming that larger colony size means more cells and larger dry weight as well (Fig. 4b). Colony areas determined from pictures taken throughout the experiment were used to determine the model parameters for the various conditions (Supplementary Table 3, Fig. 4a, c). Our assumption for the cause of the differences based on literature is twofold: one is the agar surface hydrophobicity that enables the sliding of daughter cells further from their mother[31,44], the other is increased diffusion rate in softer agar. Accordingly, 'division distance' and 'diffusion' parameters (Supplementary Table 2) were optimized to describe these two phenomena.

Simulation and experiment results both show that when colonies adjust to the changing environment, they form concentric rings of regions with different densities (Fig. 4a), furthermore simulations capture colony growth dynamics in all conditions (Fig. 4b−d).

## Predicting colony growth on plates with a nutrient gradient

Nutrients in nature are rarely available in a standard, even concentration as commonly used in the lab. We have used the model to predict colony growth in an environment with uneven nutrient distribution.

In the experiment, standard rectangular plates were filled with agar and a gradient of nutrients was established. To create a nutrient gradient, YPD was added only to 1/5th of the plate area in two different concentrations (standard and 5-times concentrated YPD agar; see details in Methods). The

**Fig. 4 | Wetness of agar plates affects the size and structure of colonies. a** Experiments (upper row, $n = 3$) and simulations (bottom row) of colony sizes and structures are shown in wet, dry, and alternating conditions after 14 days. The outer rings around the agar patties were filled with liquid YPD for 'wet' conditions and were put on a larger solid YPD plate with fresh media for 'dry' condition. Conditions were refreshed once after 7 days, either for the same or for the other condition. Simulation of similar conditions are plotted in the bottom row. Scale bar is 10 mm. **b** Cell counts, and dry weights of the colonies were determined after 14 days (color codes match the colors of **a**). **c** Agent count at the end of a 14 day simulation. **d** The area of the colonies in the various environmental conditions is shown for both the experimental (markers for each experiment separately) and one representative simulation result per experiment (lines). The vertical line shows the time of refeeding. Simulation results are scaled 55.6 pixels to 1 mm².

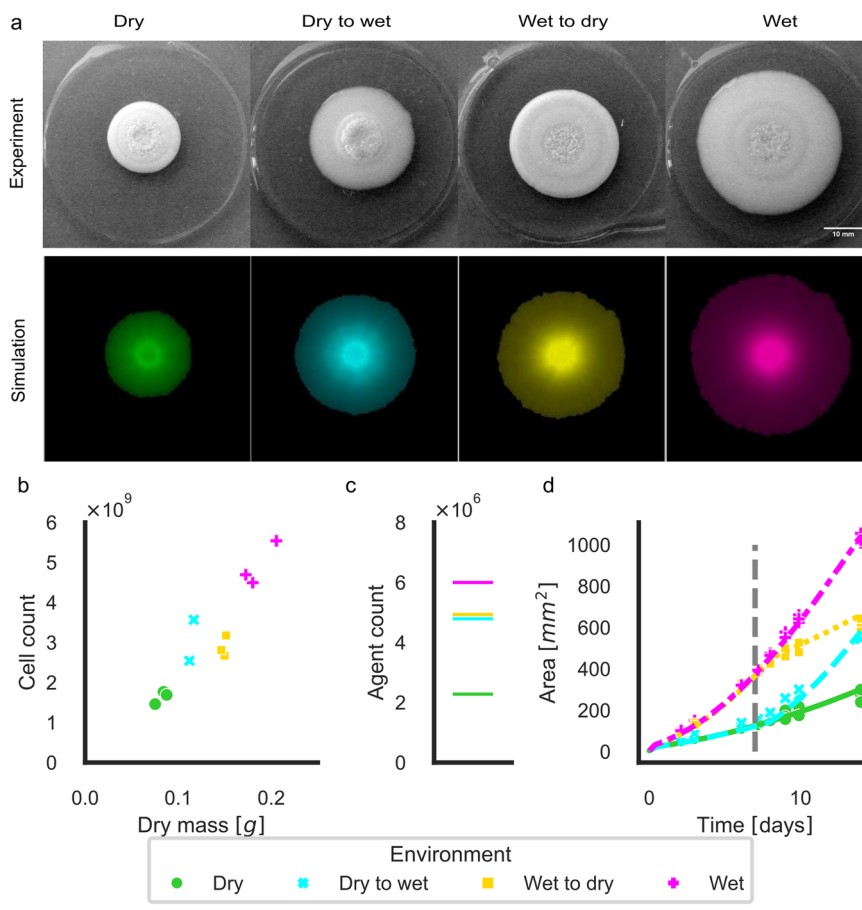

remainder of the plate was filled with agar only. This way, the 5-times concentrated plate carried the same amount of nutrients as the control plate that had YPD agar everywhere (Fig. 5). Colonies were inoculated on the lower 4/5th of the plates, where high YPD agar was not present. The inoculation pattern was designed to leave out colonies from each row while keeping 2 fully filled columns. This enables us to test how a growing colony in upper gradient might influence growth behind at a lower nutrient region. This is also another validation test for the model.

The simulation of this setup was performed by utilizing the parameter set determined by fitting the experiments with 24 colonies (Fig. 3) (exact values are shown in Supplementary Tables 2 and 3). The gradient was created by diffusion after adding the virtual nutrient to the top 1/5th of a rectangular empty simulation plane at the simulation initialization step.

Experimental results showed an early growth advantage of colonies on the control plate, especially those that were further away from the nutrient source as expected from the local scarcity of nutrients. By the 13th day (as pictured in Fig. 5a) the control plate displayed the signs of nearby colonies slowing down the growth of nearby regions, resulting in deformed colony shapes. At the same time, colony growth was visibly correlated to the nutrient gradient on the nutrient diffusion plates, with colonies expanding more towards the food source. The colonies in the bottom row were not showing any area expansion on the 1/5th normal YPD plate. Whereas on the 5-times YPD plate, all colonies were able to increase colony areas as predicted by our model. The observed colony expansions between 13 and 20 days were also fitting our model's predictions. On all plates, colony growth was observable towards 'free' areas, from which directions further nutrients became available by diffusion. This supports the power of simulation in uneven environments even though the model was not trained on data collected in such conditions. Simulations can give an additional insight into the distribution of actively dividing cells, which is difficult to measure experimentally[45]. By plotting these actively dividing cells, we can predict where subsequent growth is expected to happen (Fig. 5c).

## Predicting maximal colony size

In our parameter searching simulations, we noticed that the simulated colony could cover the whole simulation plane. During the above presented experiments, we have realized that increased agar wetness can maintain diffusion and provide nutrient availability for an extended period. This led to the hypothesis that periodic refeeding of the agar plate with liquid media can keep the colony growing unlimitedly (Fig. 6a). To test this experimentally, we created an environment where we could refeed the agar, where the colony grows (see more details in 'Methods'). In this setting we have grown a colony in a large (135 mm) Petri dish for more than 4 months (Fig. 6b). With biweekly refeed, the agar kept its wetness and fresh YPD provided nutrients for continued and renewed growth. The colony has grown over 10 cm in diameter over 4 months. Refeeding (with YPD) has caused growth rings to develop, which was also present in our computer simulations (Fig. 6a). In addition, segmental growth was observable in the experimental colony, which was only present to a lesser extent in the computer simulations. This could be due to epigenetic differentiation or differential adaptation during the lengthy experiment[46]. These are currently not incorporated in our simulator, however, our agent-based framework is suitable for incorporating stochastic changes and adaptation of cell parameters as well.

It is unclear what determines how a new inoculation nearby an already growing colony can influence the growth of the original colony. Our model predicted that a nearby colony of agents with similar parameters can inhibit the growth of the original colony in the direction of the new inoculate or slow down the growth if the new drop was on the edge of the old colony (Fig. 7a). To test this experimentally, we have inoculated fresh exponential growing cells on the edge of the growing colony and next to the colony (Fig. 7b). Both simulations and experiments show that two colonies cannot grow into each other, rather growth inhibition zone is formed between the colonies (Fig. 7). This could be the result of nutrient deprivation in this zone.

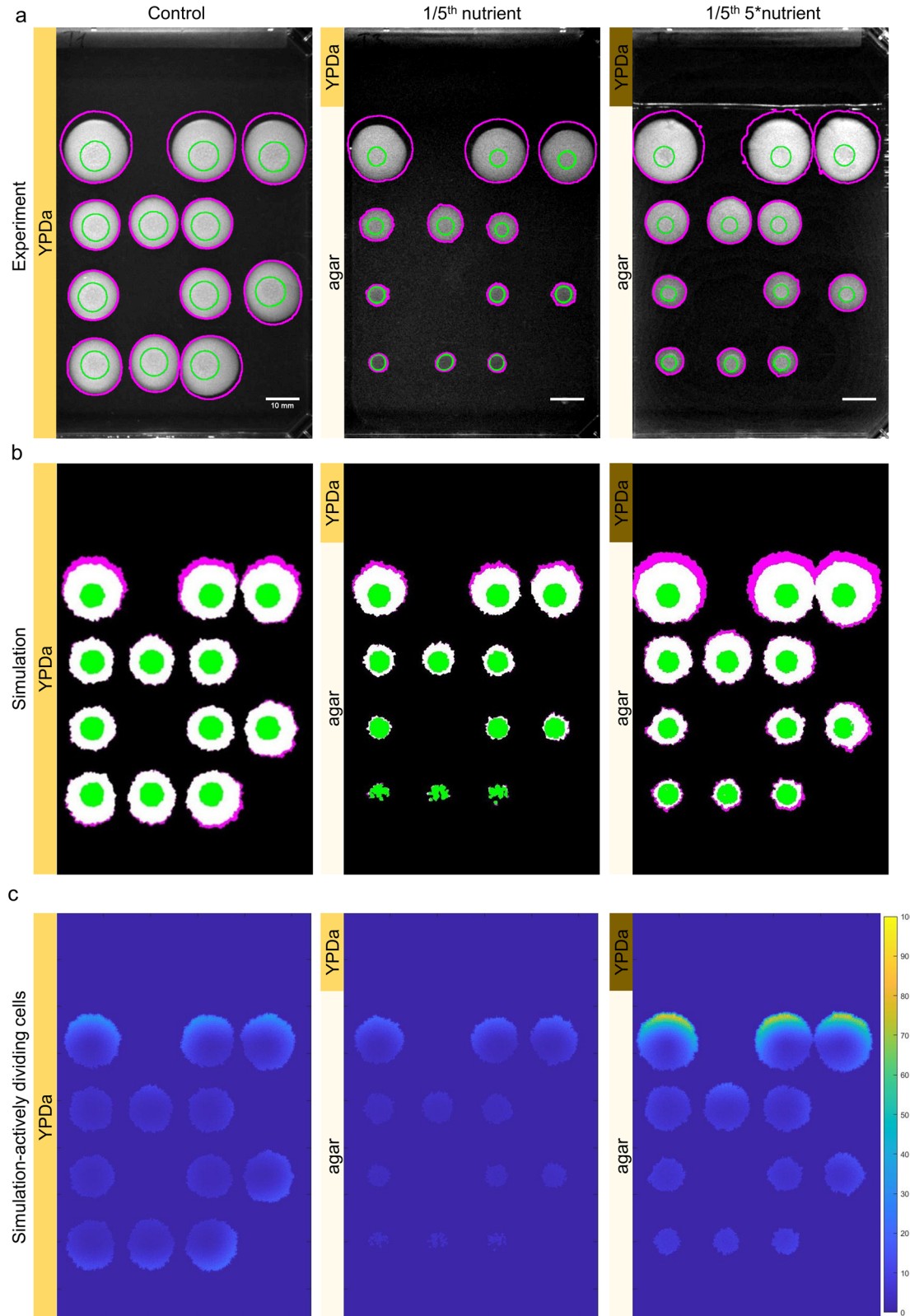

**Fig. 5 | Colony growth in uneven environment. a** Colonies were grown on OmniTray plates with even (left) or uneven (middle and right) nutrient supply for 20 days. Inoculation happened on the lower part (80% of the area) of the plate only, while nutrient was supplied on the top part (20% of the area) in the two plates on the right. Nutrient supply is indicated on the left side of the panels: control (first plate) had YPD in the whole plate, on the second plate 20% YPD and 80% only agar was used, while on the third plate, 20% 5-times concentrated YPD and 80% only agar was added. Pictures were taken on day 13 after inoculation. Green circles indicate initial colony sizes, 1−2 days after inoculation and magenta circles indicate area after 20 days, showing in which directions colonies expand between 13 and 20 days. The experiment was repeated twice with 2 plates per condition. Scale bar is 10 mm. **b** Colony area and shape are shown from the simulations of the layouts above. **c** Actively dividing cells on day 13 of the simulations, predicting directions of future growth. Agent density is indicated on a color scale.

**Fig. 6 | Growing a giant *S. cerevisiae* colony (diameter > 10 cm). a** Simulation of the colony reflecting the procedures explained on the refeed experiment. **b** A yeast colony was inoculated by a 3 µl drop on a 2% agar YPD plate on a 135 mm diameter Petri dish. The sides of the gel were removed and 8 ml of fresh liquid YPD media was added to the agar once every 2 weeks. The image was taken on day 116 after inoculation. Scale bar is 10 mm.

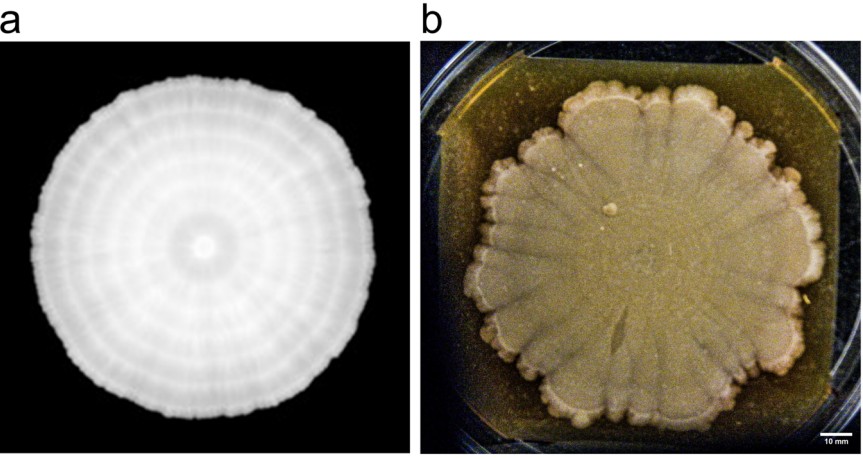

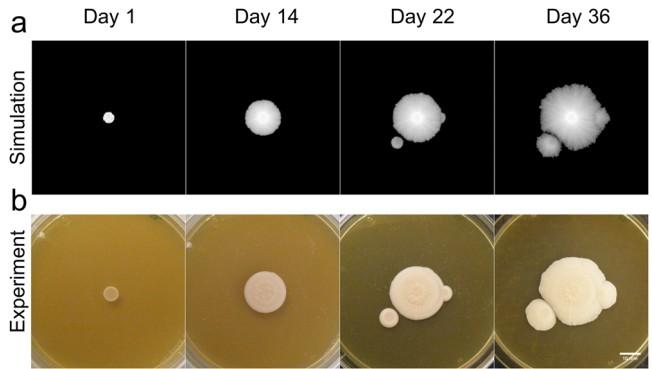

**Fig. 7 | Growing young colonies next to an aging one.** New droplets were inoculated on and next to an aging colony after 20 days of initial inoculation. Images of the colonies formed (**b**, n = 4) and their simulations (**a**) are shown on days 1, 14, 22 (2 days after second inoculation), and 36 (16 days after second inoculation). Scale bar is 10 mm.

## Discussion

Measurements on yeast strains are performed in liquid or solid media, while in their natural environment yeast cells grow in a mixture of these two conditions, most often on some wet surfaces (bark or damaged fruit). Here we aimed to investigate yeast colonies growing in time-varying and uneven environments and built an agent-based model to understand the driving and limiting factors of colony growth. We demonstrated the capability of our model of yeast colony growth to reproduce colony growth dynamics in various environmental settings. Besides our primary goal of improving, validating, and parametrizing a realistic model of yeast colony formation, we have gained valuable insights into important determining factors of colony size and development dynamics.

Both experiments and the model highlighted that the size of the initial inoculating droplet has a greater influence on colony growth than the number of cells in the initial droplet. This could be a result of local competition for available nutrients of neighboring cells in a dense droplet. Increase in cell number on a fixed drop size cannot increase much on the local proliferation of cells, but spreading out the same number of cells in a larger droplet can give them a wider area and lower density to proliferate faster.

Successful fitting of the experiments, where colonies have grown in various distances of each other (Figs. 2, 7) and where one line of the colonies was spaced with an offset from the edge of the late (Fig. 3) highlights that the model can also predict how colonies at various distances of each other and the edge of plates can grow. Yeast colonies are commonly laid out in 96, 384 dots on a plate format, and densities of up to 6144 colonies per plate were also plotted[8,9]. In most cases, statistical corrections are applied to adjust for the effect of irregularities across or between the plates[17] but our

simulation can be also used to predict how colonies would grow if competition for food is the only communication between them.

Limitations of colony growth is a recurrent and still not fully understood topic. Nutrient availability could be limiting[26], or the accumulation of an inhibitory factor (such as ammonia) produced by aging colonies[15] could limit colony size. After integrating the experimental and modeling results, we have concluded that nutrient limitation by reduced diffusion or reduced availability could be the key driving force of colony growth. The impact of local nutrient availability on colony growth is best illustrated on plates with uneven nutrient levels (Fig. 5). The 5-times concentrated YPD did provide enough nutrients to allow further away colonies to grow while diffusion from the normal YPD was only enough to feed the closer half of the colonies (Fig. 5). With these we do not rule out that ammonia or other inhibiting factors are released and slow colony growth, we only claim that all results can be fitted with a model solely focusing on nutrient diffusion.

The developed computational model is able to simulate colony growth with matching dynamics to experimental results, yet the model has several limitations. For instance, the fine structure of the colonies could be more similar between the in-vitro and in-silico experiments. This feature, and the size of the colony could be considered after implementing physical forces between the agents and between the agents and their environment (agar medium)[47,48]. There are examples for models with considerations of physical forces dominantly in the bacterial context[27,49,50] and few more specific for yeast[51,52]. Implementing these additional constraints to colony development could make our model even more realistic, however it would increase its computational requirements (or reduce the size of colony that could be simulated) and the required experimental data at the same time. Additional molecular biological mechanisms could also influence colony growth. The continuous thinning and drying of the medium could slow down colony growth. It could be introduced to the model as time and/or nutrient availability dependent division distance. Yeast cells can build extracellular matrix around themselves, and this matrix stabilizes the three-dimensional structure of colonies and affects nutrient flow inside the colonies. Cells and this matrix form together the macroscopically visible colonies[10]. The agents in the current version of the model represent a group of yeast cells and matrix components (see methods and Supplementary Table 4). Defining and visualizing the matrix itself with realistic physical properties could approximate the real macroscopic colony sizes and shapes better. It could help solving the issue that the edges of the simulated colonies are more irregular in some cases than the corresponding experimental ones (e.g: Fig. 3). In the current version, colony shapes are defined based on the simulation grid cells with agents, while in the images of in-vitro colonies only regions with a sufficient amount of cells are recognized as part of a colony. Representing the simulated colony shapes and areas with the spread of their extracellular matrix could result in smoother colony shapes. Stochastic changes could be incorporated into the model as well by defining a probability of events (e.g., cell division, death, G0 transition, nutrient

uptake) when their conditions are satisfied. Overcoming these limitations and implementing the improvements mentioned could set the directions for developing the model further.

The edges of the simulated colonies (when simulating solid agar growth) show a segmented morphology (Figs. 1, 3, 6, 7). This is getting more apparent with the limited availability of nutrients as colonies age. Pictures of the experimental colonies show this less apparently, until they grow to a real large size (Fig. 6). In the simulated colonies extracellular matrix is not present, while in real colonies these play an important role in holding cells together and keeping a sharp colony front[7,53].

We have created a software tool to simulate yeast colonies and used this model to explain the growth dynamics in numerous non-standard and non-constant environments. This helped us to understand the dynamics in non-standard environments and the way in which nearby colonies interact with each other. There is a clear need for a better understanding of microbial colony formation[54]. Since the tool is extendable to multiple strains and interaction types, potential later applications can reveal further details of the mechanisms controlling yeast colony formation in mono- and mixed cultures. Our presented experiments and model focus on a single type of yeast strain (commonly used Y55), but with strain-specific parametrization, the method can be applied to understand how interacting strains grow together. This field is already of high interest[32,55–57], and the presented combination of quantitative experiments and fitted agent-based models could help us to give a systematic insight into microbial ecology[58].

## Methods
### Strains and growth conditions
All experiments were carried out using *Saccharomyces cerevisiae* strain Y55 a/α HO::KanMX4. Media was varied according to the experiments. Standard YP (1% yeast extract (Duchefa), 2% peptone (Duchefa)) dissolved in double-distilled water was autoclaved, and glucose (Molar Chemicals) was added afterward in 2% final concentration. 5 times concentrated YPD was prepared with 5% yeast extract (Duchefa), 10% peptone (Duchefa), and glucose added in 10% final concentration. For YPD plates, agar (Duchefa) was added to YP before autoclaving; in most of our experiments 2% unless otherwise specified. Plates were poured with defined volume, 30 ml for regular (85 mm) Petri dishes, and 50 ml for OmniTray (Nunc™ OmniTray™ Single-Well Plate cat.no: 242811) unless otherwise specified. Plates were poured the day before the experiment and left to dry at room temperature for 24 h. 100 μg/μl G-418 (Duchefa) was added to prevent contamination during lengthy experiments. Cells were grown overnight in 2 ml YPD on a shaker and inoculation of a 3 μl droplet was carried out directly from the preculture onto the agar plate. After letting it absorb the droplet, colonies were grown at 30 °C over the course of the experiment. Experiments with a liquid phase were left at room temperature to prevent spills and washing away of the colonies. Single cells were separated using Singer MSM 400 micromanipulator microscope (Fig. 1b).

### Inoculating one to six colonies
Regular (85 mm) Petri dishes were poured with 30 ml soft-agar YPD (1% agar). Left to dry overnight. Yeast cells were inoculated as described above.

### 24 droplets with varied cell number and droplet size
Cell count was determined from the overnight preculture using a hemocytometer. Appropriate dilutions were prepared for each concentration according to the combination of droplet size (0.5, 1, 2, 3, 4, 5 μl) and initial cell count (1000, 2000, 4000, and 8000) pairs. I.e. preculture was diluted to $2*10^6$ cells per ml for the 0.5 μl droplet for 1000 initial cell number, this same dilution was used for the 1 μl-2000 cells spot as well. Droplets were spotted using a grid template placed under the plate. The layout was shuffled for control plates.

### Wet-dry experiment
Solidified agar plate with 25 ml YPD in a regular (85 mm) Petri dish was cut with a 50 mm circular cutter, the outer ring was removed, and 12 ml of

media (either liquid or solid) was poured around the inner disc. After 7 days remaining liquid was removed, or solid plates were cut with the method specified above, and fresh media was added to the outer ring.

### Gradient diffusion plate
50 ml agar plates were poured (2% agar dissolved in water and autoclaved) and left to dry for 24 h. The next day a straight cut was made with a blade approximately 2 cm from and parallel to the shorter edge of the OmniTray, the agar piece was removed and 10 ml YPD agar or 5-times concentrated YPD agar was added to the hole. Plates were left to dry before inoculating 3 μl droplets from overnight preculture in a 4×4 grid leaving out selected locations.

### Growing a giant colony
75 ml YPD agar was poured into a 135 mm Petri dish. On the next day, 4 slots were cut with a blade on the sides of the agar and liquid YPD was added after colony inoculation(Supplementary Fig. 5). Evaporating YPD was complemented biweekly. The plate was kept at room temperature.

### Re-inoculation experiment
To retrieve Fig. 7, new droplets (3 μl as the original) were inoculated on the side and next to an aging colony after 20 days of initial inoculation.

### Dry and wet biomass measurement
10 ml water was measured into a 50 ml tube, the colony was scraped into the tube and weight was measured before and after scraping using an analytical balance. Samples were freeze-dried to determine dry mass. Tubes containing the samples were frozen and cooled to −40 °C and introduced to a vacuum till dried for 2 days. Finally, sample tubes were measured again using an analytical balance.

### Cell counting
Selected colonies were excised from the plate (together with the underlying agar), placed in 10 ml water, vortexed, and diluted 10-fold, when necessary, then counted using a hemocytometer.

### Imaging
Two permanent photo apparatus were set up for qualitative analysis using the VWR Imager CHEMI Premium gel documentation system (Figs. 1b, 5) and one with a Fujifilm Finepix Z30 camera (Figs. 2, 4, 6, 7). For demonstrative purposes, the photos were taken with the phone's camera (Fig. 3) and stereomicroscope.

### Image analysis
The colony areas were determined using the software ImageJ. The threshold was determined using the *Auto threshold* function and the *Analyze particle tool* was used to measure colonies (colonies missed by the tool were manually outlined and measured). A custom script was used to scale up the process. Cross-sectional intensity profiles were generated using FIJI's built-in *Plot Profile* function.

Figures and statistical analysis were prepared using Python 3, Microsoft Excel and MATLAB.

### Modeling
The model was implemented in MATLAB (version: R2020b). The configuration (.csv) file contains the parameters of the simulation, its visualization, the simulated plate, the nutrient, and the strains. The model can currently handle a maximum of 24 colonies/strains and can be reached from https://github.com/CsikaszNagyLab/yeast_colony_growth_model

### General setup, scales, and simplifications of the model
The material distribution through the layers was modeled via 2-D Gaussian filtering of material values in the grid cells using the imgaussfilt function from the image processing toolbox of MATLAB. This method was chosen due to performance reasons. The standard deviation of the Gaussian

smoothing kernel is defined by the sigma parameter of the model. The two layers can have different sigma parameters. Flow (defined by equation 1 and 2) between the layers drives material equalization with a speed influenced by the flow rate parameter.

$$toplayer_{i,j} = toplayer_{i,j} + \left( \frac{deeplayer_{i,j}}{\left( \frac{init_{deep}}{init_{top}} \right)} - toplayer_{i,j} \right) \cdot \frac{flowrate}{2}$$

$$deeplayer_{i,j} = deeplayer_{i,j} - \left( \frac{deeplayer_{i,j}}{\left( \frac{init_{deep}}{init_{top}} \right)} - toplayer_{i,j} \right) \cdot \frac{flowrate}{2}$$

,where $toplayer_{i,j}$ is a grid cell of the upper nutrient layer, $deeplayer_{i,j}$ is the corresponding grid cell in the lower nutrient layer, $init_{top}$ is the initial nutrient content of the top layer, $init_{deep}$ is the initial nutrient content of the deep layer, and flowrate is the flow rate of the nutrient between the two.

Nutrient, energy levels, their uptake and consumption values are in arbitrary units but based on their ratios and assuming two hours division time for active yeast cells, one simulation step in the model corresponds to approximately 1 h. All the presented simulations use fixed boundary conditions (mimicking a Petri dish) and yeast-like growth patterns (not filamentous which is more frequent in environmental strains, not in the ones optimized for laboratory work). Signal production, time-dependent drying out of agar was not considered and the activity of G0 cells was set to zero level in order to explain experimental results with the simplest model possible concentrating on the effects of nutrient availability and environmental conditions (e.g.,: agar viscosity). The aim of the model (in its current state) is to reproduce colony areas and shapes, not the cell numbers, so the initial number of cells was one magnitude smaller than the experimental ones and the agent numbers of the final colonies also fall behind the experimentally estimated cell numbers (one agent in the model does not correspond to one yeast cell). Despite these simplifications, the model could approximate the in vivo colony sizes and growth dynamics. One to one match between cell number and agent number would have led to lengthy simulation times and high memory usage, which would not allow parameter estimation. On Supplementary table 4 we show that increase of cell numbers and rescaling growth parameters leads to similar colony sizes. We also see on Table S4, that further reduction in agent numbers leads to reduced colony size, thus our chose resolution is optimal for matching colony growth dynamics, while keeping computing time in a reasonable limit.

## Visualizations

If virtualization is enabled in the configuration file, the living agents are plotted on the top of the simulation field which is colored according to its nutrient content (the sum of the two nutrient layers). Different agent states and different agent types can be shown with separate colors (making it easier to study strain-strain interactions). After the simulation, the outputs can be visualized from multiple aspects. The number of agents separated by strains, metabolic states, and the colonies' areas can be plotted in the function of simulation steps. Final agent density patterns on the simulation grid are representing the final colony shapes. These plots can be generated for all cells (living and dead), living (active and G0), and active cells separately about the entire plate or about specific strains as well. These final images can be post-processed to be visually more comparable with the photos of in vivo colonies.

## Parameter estimation

The fitting of model parameters to experimental data was performed in MATLAB using a built-in Nelder-Mead simplex search algorithm (fminsearch)[49] initialized from a set of initial conditions producing realistic colony sizes and shapes with a reasonable number of agents (and thereby reasonable simulation times). The minimized objective function was the sum of squared differences between simulated and experimental colony areas (expressed in the number of pixels) as described in equation 3.

$$diff = \sum_{i=1}^{n} \left( sim_i - ref_i \right)^2$$

,where diff is sum of squared difference, n is the number of experimental points, $ref_i$-s are the experimental colony sizes and $sim_i$-s are the corresponding simulated colony areas (expressed in pixels).

Experimental measurement data was sparser than the simulated one (typically maximum one photo was taken about the colonies each day while one simulation step in the model represents approximately one hour), so the simulated data points were sampled, and the temporally matching point pairs were used in the fitting. Fitted parameters consist of cell division distances, nutrient diffusion rates, initial nutrient levels, and initial colony sizes.

## Statistics and reproducibility

Experiments included typically 3−6 replicates repeated at least once. Actual sample sizes are indicated in the figure legends.

## Reporting summary

Further information on research design is available in the Nature Portfolio Reporting Summary linked to this article.

## Data availability

All data supporting the figures are stored together with their codes on the Github repository https://github.com/CsikaszNagyLab/yeast_colony_growth_model (https://doi.org/10.5281/zenodo.10848933)[59].

## Code availability

Link to the code, to the configuration files (with the parameters used for the simulations of the manuscript), and to the instructions of running the model: https://github.com/CsikaszNagyLab/yeast_colony_growth_model (https://doi.org/10.5281/zenodo.10848933)[59].

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

## Acknowledgements

We acknowledge support by the Hungarian National Research, Development and Innovation Office (NKFI/NRDI) through the Hungarian Scientific Research Fund (OTKA-K20-134489) and the Thematic Excellence Programe grant no. TKP2021-EGA-42. We appreciate the valuable comments on our manuscript by Gábor Balázsi, Andrea Ciliberto and Irene Stefanini.

## Author contributions

T.G., B.P., C.I.P., and A.C-N. conceived and planned the experiments. T.G., B.P., H.S. and C.I.P. carried out the experiments. T.G., J.J. and A.C-N. planned and carried out the simulations. All contributed to the interpretation of the results. T.G., J.J., and A.C-N. wrote the manuscript. All authors provided critical feedback and helped shape the research, analysis, and manuscript.

## Funding

## Competing interests

The authors declare no competing interests.
