## [Peer Review File · Communications Biology]

Reviewers' comments:

Reviewer #1 (Remarks to the Author):

In this work, the authors study the growth of yeast colonies under diverse conditions. In particular, they study the impact of initial seeding, nutrient intake and humidity, along with temporal and spatial perturbations, on the colony growth. For this purpose, they make use of experiments and agent-based simulations fitted to the experiments to make a predictive model of yeast growth.

I highly appreciate the efforts to bridge the gap between experiments and simulations by fitting their models to actual data.

Regarding the biological originality of the work. To my opinion, the most biologically main result of the work is stated by the paper in lines 478-480 of the discussion: "After integrating the experimental and modeling results, we have concluded that nutrient limitation by reduced diffusion or reduced availability could be the key driving force of colony growth." Limitation of nutrient availability because of reduced transport (diffusion) or environmental limitations (availability), is a well known phenomenon in colony growth.

The differential contribution of the work over existing data is the inclusion of the effects' of wetness, temporal and spatial inhomogeneities in the growth conditions. The results are nice, but not unexpected. My wants to remark the difficulty to understand how important is this additional contribution since the article suffers from several argumentative flaws, limited contextualization of the work, missing information and unintuitive simulation conditions which are not properly justified. While I encourage the authors to pursue the publication of the article (here or somewhere else), it considers that the manuscript and some analysis performed in it require extensive revision.

In the following, I will try to point out all the major issues of the article, according to my opinion. For helping with this task, I provide an added file with a line enumeration to help to navigate the comments.

Major concerns

Unfocused and incomplete introduction. To my point of view, the introduction is very confusing. By the end of the section, the goals of the work are not well stated, and the message of the authors only becomes clear after reading the entire paper. The introduction shows potential lack of references and adequate contextualization of the work with the existing literature, justifying the limitations of the previous work and stating the contribution of this work. In the following, I will refer to several issues observed to substantiate my claims:

I see an unclear formulation of the limitations of previous work. Here I will point out some examples.

"Growing yeast or bacterial colonies on semi-solid surfaces is an everyday task in most laboratories. Still, several details of the mechanism and dynamics of colony growth are unknown. The first detailed investigation..." (lines 29-31)

The article claims that there are details missing on the mechanisms of colony growth. Immediately after, it is stated that in 1970 Meunier and Choder made a detailed investigation on the phases of colony growth. The article does not mention what are the details missing, nor states the limitations of a work made already fifty years ago.

Second, the introduction has many sentences with vague meaning, that seem to point (if I interpreted them correctly) at the original contribution of the work, but this is never really specified. The introduction also seems to lack extensive literature revision.

“Growing yeast or bacterial colonies on semi-solid surfaces is an everyday task in most laboratories. Still, several details of the mechanism and dynamics of colony growth are unknown. The first detailed investigation...” (lines 29-31)

“Growing yeast colonies and using colony size as an indicator of fitness is a daily routine in many research labs. Although, many details on the mechanism and dynamics of colony growth are less understood. It is also not fully clear what serves as a limiting factor of colony expansion.” (lines 37-40...)

The second statement above mentions the unknown limiting factor of colony expansion. However, a very rich literature has shown evidence of the factors influencing colony growth, and even studied the complex mechanistic strategies that the colonies use to grow under conditions such as limited nutrient sharing. Here I leave just some examples in the context of bacterial colonies and biofilms.

<https://doi.org/10.1038/nature15709>

<https://doi.org/10.1016/j.bpj.2022.11.2853>

DOI: 10.1126/science.aah4204

References are also missing in the context of mathematical modeling.

“Models were also derived to capture colony growth dynamics. Some mathematical models focus on factors controlling yeast colony expansion, and recently models also started to study how the structure of the colonies might vary. (Just one cite)” (lines 48-50)

The articles proposed above include all modelling on the corresponding growing colonies and showing complex time and spatial behaviors. The article would benefit from a more extensive literature search and/or and stronger justification of why not to include a big part of the existing references on this topic. Maybe, if what they mean is lack of modelling in yeast, specifically, they should reinforce this.

One of the most clearly stated problems in the introduction is the lack of study of the dynamical changes of environmental conditions. However, in the end, the presented work does not include any fluctuating condition experiments (understood by me as temporal fluctuating conditions). Such paragraph is misleading to the actual claims of the paper.

“Dynamical changes in environmental conditions are the default state in nature and yeasts have evolved

and adapted to this situation. On the contrary, in laboratory experiments, we tend to minimize the dynamic variations in conditions (media, temperature, humidity) mostly for standardization and reproducibility. There are only a few examples where researchers considered alternation in an environmental factor, like temperature.” (lines 42-46)

In this case, the authors consider that the paper has indeed studied dynamics in the environment; it would be appreciated a deeper explanation of what is the meaning of dynamics in the context of their work, both here and in the corresponding results section.

By the end of the introduction, the authors have not reached the specific goals of their study (3), how this study is different from the other existing in the literature (1), and confused the reader with unjustified claims, many of which are not present in the rest of the study (2).

“(1) The dynamics, structure, and nutrient dependence of yeast colony growth were tested on various strains. (2) Here, we have understood that communication might be present between colonies, environmental conditions have effects on growth, mutations can affect colony morphology, diffusion can limit growth, and colony morphology changes can be captured by mathematical models as well. (3) Here, we combine quantitative colony measurements in various conditions with agent-based modeling to reveal the dynamical features and limiting factors of colony growth.” (lines 55-61)

Results: Agent-based model justification. There is in general a clear explanation of the agent based model implemented, both in the results section and in the supplementary table describing all the parameters. However, this reviewer finds some information missing in the manuscript to completely understand the simulator, as well as some biological justification, and a discussion of simulation limitations that should be addressed for a complete description of the section.

Asymmetric division. The division algorithm described is asymmetric, retaining one cell as the mother and the other as a daughter. This simulation decision is far from obvious, and a biological justification should be provided. If this decision is given because of the asymmetric division of budding yeast strains as the budding yeast, this should be mentioned somewhere.

“Cell division happens in two dimensions: the daughter cell gets a random position at a certain distance from the mother cell (the daughter cell has the same properties as the mother cell except for its position and energy level).” (lines 79-82)

Distance to the mother. From the above description, it seems the model generates a daughter that may be several grid steps apart from the mother. However, the middle grids the daughter will be jumping over, may contain cells. To me, this approach is extremely awkward, as “it would look like you are generating matter at a distance”. A deep justification of this unintuitive simulation approach should be addressed and their implications to the simulations addressed.

The paper should mention the topology of the simulation grid (square, hexagonal...).

The diffusion algorithm through the layers is not provided.

The model considers a grid with two layers of nutrients. However, the experimental setup is just an agar medium. To me, there is not a justification of why it is necessary, or how biologically reasonable it is, to model the system with two layers. A justification of this modelling decision should be provided.

“The simulation plane consists of two layers of nutrients, the upper one can be consumed by the cells directly while the lower one is used to recharge the upper one.” (lines 93-95)

Results: Testing the model.

The paper mentions that they fit by area and shape, but in the end the quantitative fittings are just done by area.

“First, we have tested the model on fitting colony sizes and shapes formed after various inoculations.” (lines 138-139)

The article claims that the simulations reproduce the “volcano shape” also present in the simulations. However, I cannot see that from the 2D images. I would suggest showing a figure with the density of cells along a diameter (Z axis) outline to provide evidence of this.

“Single-cell colonies and colonies inoculated from a droplet show distinct colony shapes. When considering local cell density as a third dimension, the simulations represent well the conical shape of a single cell colony and the volcano-like shapes experienced when inoculating with liquid droplets. The difference in the crater size depending on the inoculation droplet size is also apparent (Fig. 1b).” (lines 139-142)

I also does not see clearly if simulations represent the shape or if this is the initial condition of the simulations (inoculation condition). A clarification would be appreciated.

Results: Growth dynamics parametrization

The model has over thirty parameters, but only six are fitted. While some of them are fixed and explained in Supplementary Table 2, most are not discussed. A more extended discussion on why the other parameters are not fitted, even if it is because they are irrelevant to the final outcome, should be addressed.

“The area over time was used to determine the following parameters of the model: cell division distances, nutrient diffusion rates, initial nutrient levels, and initial colony sizes.” (lines 160-161)

Results: Understanding initial conditions of inoculation.

The paper should state what the predictions are. Everything looks like a fitting to the data.

“We have tested our model to predict this unevenness for cases when statistics are not feasible or significant.” (lines 184-185)

It is not clear if the parameters are fitted for each colony separately or all of them together. Clarification on the fitting process should be made.

“In order to mimic the conditions of regular high throughput OmiTray experiments when colonies grow close to each other for a shorter time, an agar layer was defined with high initial nutrient level, which is constantly consumed by cells at all spots. Diffusion values, cell division distances and initial nutrient amounts were used for fitting the colony sizes to the experimental data.” (lines 197-201)

Figure 3 shows that experimental colonies are smooth, while simulations have a more anomalous contour. While the paper discusses the unevenness in the last section of the results when it happens in the other direction, it does not mention anything in this one. This reviewer thinks that it may be a problem of the order of magnitude difference between simulations and experimental sizes. This should be clarified.

Results: How agar wetness affects colony size and structure?

I consider this section very nice but finds it unfortunate that the fitting process is not clearly explained, and some important points are left unanswered.

How is the data in Figure 4c fitted? Are the Dry and Wet experiments fitted, and the other two predicted using this fitting? Are all four conditions fitted? In the latter case, why?

How do the fitted parameters adjust to the changing environments? The change in fitted parameters between conditions should be shown somewhere and commented upon.

The fitting only changes division distance and diffusion, however Figure 4b suggests that there is more proliferation of cells. Why has this not been fitted or predicted? If not included in a new fitting, it would be relevant to add some comments about it. This reviewer feels a bit awkward that this information appears in a Figure with all the other images being fittings, except from this one.

“Accordingly, ‘division distance’ and ‘diffusion’ parameters were optimized to describe these two phenomena. Simulation and experiment results both show that when colonies adjust to the changing environment, they form concentric rings of regions with different densities (Fig 4a), furthermore simulations capture colony growth dynamics in all conditions (Fig 4c).” (lines 256-263)

Results: Predictions with the model

The first sentence is very vague, and no further information is included. The paper should include a detailed description on what parameters have been used from all the above-mentioned fittings, how the simulated experiment has been set and the predictions clearly stated.

“Nutrients are rarely available in a standard, even concentration as commonly used in the lab. We have used the model and parameters optimized in the above-described experiments to predict colony growth in an environment with uneven nutrient distribution.” (lines 323-325)

This introduction, again, is vague. It should be further detailed. If it is a prediction, they should state how

they performed the simulations again.

“In our simulations, we realized that, when parameters of nutrient availability are perturbed, then the simulated colony could cover the whole simulation plane, independent of the size of the plane. This inspired us to test if we can grow a giant colony, which can cover the whole area of a Petri dish.” (lines 401-405)

Discussion:

The following statement was not a prediction, according to the article, but a fitted property after experiments. Here you have changed the logical order of the results.

“Our model suggested and experiments confirmed...” (line 458)

The paper does not state the efficiency of simulation capabilities of their software. Along with the comments on the code availability and documentation mentioned further below, I consider the following claims overstatements:

“We have created a powerful tool to simulate yeast colonies and used this model to explain the growth dynamics in numerous non-standard and non-constant environments.” (lines 486-487)

“Since the tool is easily extendable to multiple strains and interaction types, potential later applications can reveal further details of the mechanisms controlling yeast colony formation in mono- and mixed cultures.” (lines 489-492)

Minor concerns

The paper goal does not seem to be an exploratory analysis of many conditions. The work already selects a set of factors to study, names seeding, humidity and nutrient intake (excluding many other relevant factors such as temperature, other nutrients' intake, competition...) so the statement that one of the goals of the article is to identify the crucial factors looks like an overstatement of the achievements of the article. When using the term “including”, it gives the impression that the article studied other factors that the work shown to be irrelevant to the colony growth; but the article just models and studies the mentioned factors.

“To identify the crucial factors of colony growth dynamics, we have developed and parameterized a quantitative agent-based model of yeast colony growth by synergizing mathematical modeling with laboratory experiments.” (line 13-16)

“Through colony growth experiments and model fitting, we demonstrate the influence of various environmental factors, including moisture, nutrient availability, and initial colony inoculation conditions.” (line 17-20)

It is not very clear what the authors meant with this sentence. Was it meant to be a “However”, maybe?

“Growing yeast colonies and using colony size as an indicator of fitness is a daily routine in many research labs. Although, many details on the mechanism and dynamics of colony growth are less understood.” (lines 37-39)

It may have been nice to quantify the shape parameter, or at least comment how it could have been used quantitatively.

“One can observe that colony shapes change from circular to a less and less circular shape due to the proximity of neighboring colonies and the edge of the plate. This feature was also present in our simulations (Fig. 2b).” (Lines 153-157)

Although this reviewer has read the materials section and seen the cell counting process; it is still unclear to them how you control cell number at the seeding time. A further clarification would be nice.

“Standard rectangular plate format (OmiTray) was used to investigate two initial inoculation parameters: droplet size and cell number.” (lines 186-187)

Figure 3 is somehow confusing, as the a and b subsections show initial droplet x initial cell count, but subcaption c shows time x area. Maybe putting the a and b axis labels in the top and right sides of the figure may help the reader to see the equivalence.

There is no evidence in the distribution of active cells since, as mentioned in the same paper, there are no observables of this distribution. The authors should reconsider if this is a prediction of the model, as the authors state. From the sentence, I interpret it as a known true fact.

“On all plates, colony growth was observable towards ‘free’ areas, from which directions further nutrients became available by diffusion. This supports the power of simulation in uneven environments, but simulation can give an additional insight that is difficult to measure experimentally, which is the distribution of active cells. By plotting actively dividing cells, we can predict where growth is expected to happen (Fig. 5c).” (lines 340-345)

Data availability.

I have not been able to find the data of the article, nor is it mentioned anywhere if it will be published anywhere at some point.

Code availability.

The repository is public and well organized. However, the documentation is very limited and only contains a minimal set of instructions to run the simulations performed in the article. The code for the data analysis is not provided in the repository. I am concerned about the claims made by the paper that this software could be used to perform custom analysis and extension, as no information on how to use the code outside the provided simulations is given. I am also concerned about the programming language being used. While the code of the article is publicly available, MATLAB is a proprietary software that is not common in many laboratories nowadays, and this limit the accessibility of the results to a

large audience. Analyzing the code script, I see no technical burdens to implement this code in an open language such as Python, C or Julia. This reviewer is aware of the complexities of programming and is not asking for migrating the code to a different language for publication; but wants to remark, for future work and for the sake of code transparency in science, that the authors reconsider the programming language that they choose to use.

Reviewer #2 (Remarks to the Author):

In this paper, the authors presented a modeling approach to analyze different environmental factors on yeast colony growth. The growth of yeast cells in a defined Petri dish environment is a spatio-temporally complex. The paper has a simple algorithm that connects several factors including nutrient availability and gradient, and inoculation size to colony morphology, and offer good insights on how these factors shape the morphology all together.

My comments and suggestions for revisions are as below:

(1) Fig 1a: The manuscript currently contains no mathematical equations, nor detailed explanations on how parameters are fitted and trained. It is not clear how different parameters are connected, and why the choice of unitless parameters, and how certain threshold parameters such as cell death and division were chosen rationally (whether it's based on biologically realism). The authors also mentioned about the possibility of adding stochastic cell birth-death processes. It's worthwhile to explain how. Given that modeling is the major innovation in this paper, I would suggest a major revision on this part.

(2) Fig 1b: Based on the picture it doesn't seem like the model simulation recreated the volcano crater size. Suggest to plot the colony in cross-sectional view.

(3) Fig 3: Suggest to provide a quantitative measurement of the effects of initial cell count and droplet size. It's not clear the degree of interactions between the two parameters on the colony area.

(4) It's confusing how agent numbers correlate with the cell numbers, and why there's a magnitude of difference between the number of cells/agents in simulation and in experiments. Why is the model not being able to fit this? If the author is predicting the effect of initial cell number to the colony area in Fig 3, the cell number parameter seems to be very important in the model.

(5) Fig 4: top panel should be removed. It is redundant.

(6) Fig 5: The arrangement of colonies seem arbitrary. Explain.

(7) Fig 6: How was the media replenished? Does the formation of "rings" correlate with the media renewal? In Fig 6c, the authors added a drop on and next to an aging colony, but didn't explain the rationale of this experiment nor the interpretation of it.

We respond to each comment of the referees with blue colors below.

Reviewer #1 (Remarks to the Author):

Summary

In this work, the authors study the growth of yeast colonies under diverse conditions. In particular, they study the impact of initial seeding, nutrient intake and humidity, along with temporal and spatial perturbations, on the colony growth. For this purpose, they make use of experiments and agent-based simulations fitted to the experiments to make a predictive model of yeast growth.

I highly appreciate the efforts to bridge the gap between experiments and simulations by fitting their models to actual data.

Regarding the biological originality of the work. To my opinion, the most biologically main result of the work is stated by the paper in lines 478-480 of the discussion: “After integrating the experimental and modeling results, we have concluded that nutrient limitation by reduced diffusion or reduced availability could be the key driving force of colony growth.” Limitation of nutrient availability because of reduced transport (diffusion) or environmental limitations (availability), is a well known phenomenon in colony growth.

The differential contribution of the work over existing data is the inclusion of the effects' of wetness, temporal and spatial inhomogeneities in the growth conditions. The results are nice, but not unexpected. My wants to remark the difficulty to understand how important is this additional contribution since the article suffers from several argumentative flaws, limited contextualization of the work, missing information and unintuitive simulation conditions which are not properly justified. While I encourage the authors to pursue the publication of the article (here or somewhere else), it considers that the manuscript and some analysis performed in it require extensive revision.

We thank the referee for pointing out the main findings of our manuscript. We have thoroughly updated the manuscript to make clear how our contributions fit and extend earlier findings in the field.

In the following, I will try to point out all the major issues of the article, according to my opinion. For helping with this task, I provide an added file with a line enumeration to help to navigate the comments.

Major concerns

Unfocused and incomplete introduction. To my point of view, the introduction is very confusing. By the end of the section, the goals of the work are not well stated, and the message of the authors only becomes clear after reading the entire paper. The introduction shows potential lack of references and adequate contextualization of the work with the existing literature, justifying the limitations of the previous work and stating the contribution of this work. In the following, I will refer to several issues observed to substantiate my claims:

- I see an unclear formulation of the limitations of previous work. Here I will point out some examples.

“Growing yeast or bacterial colonies on semi-solid surfaces is an everyday task in most laboratories. Still, several details of the mechanism and dynamics of colony growth are unknown. The first detailed investigation...” (lines 29-31)

The article claims that there are details missing on the mechanisms of colony growth. Immediately after, it is stated that in 1970 Meunier and Choder made a detailed investigation on the phases of colony growth. The article does not mention what are the details missing, nor states the limitations of a work made already fifty years ago.

We have rewritten most of the introduction to better capture the existing literature and the history of investigating microbial colony growth both experimentally and by models. We have also pointed out the major limitations in our knowledge and explain how we expand on these.

In the updated introduction we go far beyond works by Meunier and Choder (1999) and Gray and Kirwan (1974) by presenting the earlier and later papers on general microbial and specifically on yeast colonies. We also highlight that most mathematical models considered only the diameter of colonies, a few discussed their shape and only a limited number considered individual cells and their interactions from both experimental and (agent-based) modeling direction. The missing point from the literature is what really limits the growth of colonies and our current knowledge still lacks quantitative details on how growth of colonies depend on diffusibility of media in the agar and how initial layouts of cells can influence colony growth. These are the major points we address in this manuscript. (lines 28-32; 39-56)

All these are explained in detail in the revised introduction.

- Second, the introduction has many sentences with vague meaning, that seem to point (if I interpreted them correctly) at the original contribution of the work, but this is never really specified. The introduction also seems to lack extensive literature revision.

“Growing yeast or bacterial colonies on semi-solid surfaces is an everyday task in most laboratories. Still, several details of the mechanism and dynamics of colony growth are unknown. The first detailed investigation...” (lines 29-31)

“Growing yeast colonies and using colony size as an indicator of fitness is a daily routine in many research labs. Although, many details on the mechanism and dynamics of colony growth are less understood. It is also not fully clear what serves as a limiting factor of colony expansion.” (lines 37-40...)

The second statement above mentions the unknown limiting factor of colony expansion. However, a very rich literature has shown evidence of the factors

influencing colony growth, and even studied the complex mechanistic strategies that the colonies use to grow under conditions such as limited nutrient sharing. Here I leave just some examples in the context of bacterial colonies and biofilms.

<https://doi.org/10.1038/nature15709>

<https://doi.org/10.1016/j.bpj.2022.11.2853>

[DOI: 10.1126/science.aah4204](https://doi.org/10.1126/science.aah4204)

References are also missing in the context of mathematical modeling.

“Models were also derived to capture colony growth dynamics. Some mathematical models focus on factors controlling yeast colony expansion, and recently models also started to study how the structure of the colonies might vary. (Just one cite)” (lines 48-50)

The articles proposed above include all modelling on the corresponding growing colonies and showing complex time and spatial behaviors. The article would benefit from a more extensive literature search and/or and stronger justification of why not to include a big part of the existing references on this topic. Maybe, if what they mean is lack of modelling in yeast, specifically, they should reinforce this.

As explained in the above point, we have included a more extensive literature review, including the mentioned references and many others. The new introduction also discusses the literature of bacterial colony growth models.(lines 44-56)

- One of the most clearly stated problems in the introduction is the lack of study of the dynamical changes of environmental conditions. However, in the end, the presented work does not include any fluctuating condition experiments (understood by me as temporal fluctuating conditions). Such paragraph is misleading to the actual claims of the paper.

“Dynamical changes in environmental conditions are the default state in nature and yeasts have evolved and adapted to this situation. On the contrary, in laboratory experiments, we tend to minimize the dynamic variations in conditions (media, temperature, humidity) mostly for standardization and reproducibility. There are only a few examples where researchers considered alternation in an environmental factor, like temperature.” (lines 42-46)

In this case, the authors consider that the paper has indeed studied dynamics in the environment; it would be appreciated a deeper explanation of what is the meaning of dynamics in the context of their work, both here and in the corresponding results section.

In our work we followed the dynamical loss of nutrients from the media (figures 1-3, 5) and we have also investigated experimentally and by model as well what happens if we add extra nutrients once (Fig. 4) or multiple times (Fig. 6a,b) to the

agar where the colony grows. We have also checked how a colony changes its growth when new cells are added next to the colony after 20 days (Fig. 7).

We have rephrased this section of the introduction to tone down the dynamical aspects of our research, but we explained these now in the discussion to clarify what we mean by dynamical perturbations of colony growth (lines 366-370, 428-430).

- By the end of the introduction, the authors have not reached the specific goals of their study (3), how this study is different from the other existing in the literature (1), and confused the reader with unjustified claims, many of which are not present in the rest of the study (2).

“(1) The dynamics, structure, and nutrient dependence of yeast colony growth were tested on various strains. (2) Here, we have understood that communication might be present between colonies, environmental conditions have effects on growth, mutations can affect colony morphology, diffusion can limit growth, and colony morphology changes can be captured by mathematical models as well. (3) Here, we combine quantitative colony measurements in various conditions with agent-based modeling to reveal the dynamical features and limiting factors of colony growth.”
(lines 55-61)

We hope that with the above mentioned revisions the goals of the study are clear now and it is also properly stated what we add as new results to the existing literature. We have also tried to better streamline the introduction to focus on what our study is leading to. (lines 62-67)

Results: Agent-based model justification. There is in general a clear explanation of the agent based model implemented, both in the results section and in the supplementary table describing all the parameters. However, this reviewer finds some information missing in the manuscript to completely understand the simulator, as well as some biological justification, and a discussion of simulation limitations that should be addressed for a complete description of the section.

- Asymmetric division. The division algorithm described is asymmetric, retaining one cell as the mother and the other as a daughter. This simulation decision is far from obvious, and a biological justification should be provided. If this decision is given because of the asymmetric division of budding yeast strains as the budding yeast, this should be mentioned somewhere.

“Cell division happens in two dimensions: the daughter cell gets a random position at a certain distance from the mother cell (the daughter cell has the same properties as the mother cell except for its position and energy level).” (lines 79-82)

Yeast cell division is an asymmetric process. This biological knowledge led us to the choice of asymmetric division algorithm. Hence no active movement is possible, and the daughter cells are smaller, we have assumed the mother cell to remain at the original location and the daughter cell to pursue a new location. This is a simplification as physical forces are excluded. In a future version of the model this will also enable us to follow the aging of mother cells, which can divide only approximately 30 times before dying (Denoth Lippuner, A., Julou, T., & Barral, Y. (2014). Budding yeast as a model organism to study the effects of age. *FEMS Microbiology Reviews*, 38(2), 300-325.).

The smaller size of the budding yeast daughter cell is modeled by its initial energy content, which can be smaller than the remaining energy content of the mother cell. (lines 86-87)

“This asymmetric cell division reflects the typical behavior of budding yeast”

- Distance to the mother. From the above description, it seems the model generates a daughter that may be several grid steps apart from the mother. However, the middle grids the daughter will be jumping over, may contain cells. To me, this approach is extremely awkward, as “it would look like you are generating matter at a distance”. A deep justification of this unintuitive simulation approach should be addressed and their implications to the simulations addressed.

Agents do not have explicitly defined physical extent in the model. With the selected parameters daughter agents are typically positioned to neighboring grid cells of their mother. If in a special case (what we haven't seen, but theoretically possible) a daughter cell is generated several grid steps apart from the mother cell, it indicates a mother cell size of several grid units, which we did not see with the current parameter settings.

Theoretically overlapping agents can be interpreted as agents on the top of each other. The model does not use direct three dimensional coordinates, but can describe more complex structures (not only single cell layers) in this way (lines 96-104).

- The paper should mention the topology of the simulation grid (square, hexagonal...).

Square grids are used. We have included this explicitly in the paper. (line 93)

“The agar plate is simulated as a two-layer square grid plane...”

- The diffusion algorithm through the layers is not provided.

The material distribution through the layers was modeled via 2-D Gaussian filtering of material values in the grid cells using the `imgaussfilt` function from the image processing toolbox of MATLAB. This method was chosen due to performance reasons. The standard deviation of the Gaussian smoothing kernel is defined by the sigma parameter of the model. The two layers can have different sigma parameters (which are given in Table S2 and S3). Flow between the layers drives material equalization with a speed influenced by the flow rate parameter. (lines 513-523)

- The model considers a grid with two layers of nutrients. However, the experimental setup is just an agar medium. To me, there is not a justification of why it is necessary, or how biologically reasonable it is, to model the system with two layers. A justification of this modelling decision should be provided.

“The simulation plane consists of two layers of nutrients, the upper one can be consumed by the cells directly while the lower one is used to recharge the upper one.” (lines 93-95)

The nutrient layer is 2-dimensional in our simulations. However the depth and diffusion in lower layers of the agar has an important role in colony growth (as this is demonstrated in Figure 5). Since there is no 3rd dimension in our model, we have included just a second layer of nutrients. This implementation simplifies (and speeds up) the simulation of the depth of the nutrient agar. The biological consideration behind this can be interpreted as the discretisation of the nutrient. Upper layer consists of the amount of nutrients available for the cells within one iteration. This reasoning is properly explained in the methods section of the updated manuscript. (lines 94-98)

Results: Testing the model.

- The paper mentions that they fit by area and shape, but in the end the quantitative fittings are just done by area.

“First, we have tested the model on fitting colony sizes and shapes formed after various inoculations.” (lines 138-139)

The shape was primarily observed qualitatively, but quantitative cross-sectional plots of the colonies were created for the resubmission presented in Figure 1 and Supplementary figure 1 now.

Figure 1 has been updated with the cross-sectional intensity plots and 2 days old pictures were changed for 8 days old pictures as below:

Supplementary figure 1 shows the asymmetric radial growth depending on the number of colonies on the plate (supplementing Figure 2):

of drops
on the
plate

1

2

3

4

6

- The article claims that the simulations reproduce the “volcano shape” also present in the simulations. However, I cannot see that from the 2D images. I would suggest showing a figure with the density of cells along a diameter (Z axis) outline to provide evidence of this.

“Single-cell colonies and colonies inoculated from a droplet show distinct colony shapes. When considering local cell density as a third dimension, the simulations represent well the conical shape of a single cell colony and the volcano-like shapes experienced when inoculating with liquid droplets. The difference in the crater size depending on the inoculation droplet size is also apparent (Fig. 1b).” (lines 139-142)

I also does not see clearly if simulations represent the shape or if this is the initial condition of the simulations (inoculation condition). A clarification would be appreciated.

We thank the referee for this suggestion. We have included intensity plots showing the picture intensities along the diameter for both the experimental and simulated colonies on Figure 1 b,c and Supplementary fig 1. On Figure 1 b,c: for better illustration we have changed the 2 days old pictures for 8 days old pictures where the volcano shape is more evident.

The methods section was complemented with the used methodology (lines 504-505):

Cross-sectional intensity profiles were generated using FIJI’s built-in Plot Profile function.

Results: Growth dynamics parametrization

- The model has over thirty parameters, but only six are fitted. While some of them are fixed and explained in Supplementary Table 2, most are not discussed. A more extended discussion on why the other parameters are not fitted, even if it is because they are irrelevant to the final outcome, should be addressed.

“The area over time was used to determine the following parameters of the model: cell division distances, nutrient diffusion rates, initial nutrient levels, and initial colony sizes.” (lines 160-161)

Supplementary Table 2 has been updated to contain a detailed description of all parameters and the selected values for each. Supplementary table 3 contains parameters, which were changed for the simulations of individual experiments. The considerations behind using the chosen parameters for the fitting were added to the “Growth dynamic parameterization” section. (lines 145-172, Supplementary Table 3: lines 753-757)

Supplementary table 2: Input parameters and output descriptions of the model.
Green: temporal parameters; yellow: spatial parameters of the environment or the

agent; red: parameters defining material distribution in the environment; blue: cell metabolism and energetics related parameters; bold: fitted or fine-tuned parameters. Input parameters are defined in a .csv file. The simulations produce outputs for on-the-fly data visualizations, but also save .mat and .tsv data files for post processing.

Parameter name	Used value(s)	Description
General and visualization parameters		
Iteration number*	228 - 2160	length of the simulation (in simulation steps) 1 step~1 hour with the default settings
Time step	1	time resolution
Draw	0,1	1: visualizations during the simulation are on 0: no visualizations during the simulation
Visualization steps	20	frequency of updating the graphical output during the simulation (x indicates update in every xth steps)
Environmental (plate, agar and nutrient related) parameters		
Plate x*	620 - 900	x dimension of the plate (in grid cells)
Plate y*	620 - 900	y dimension of the plate (in grid cells) 1 grid cell ~ 55.6 mm ² with default settings
Boundary condition	1	0: periodic boundary condition 1: fixed boundary condition (used in the presented work)
Sigma1*	30 – 80	diffusion parameter of the top nutrient layer
Sigma2*	25 - 66.67	diffusion parameter of the deep/lower nutrient layer
Flow rate	0.01	nutrient exchange rate between the two nutrient layers
Initial nutrient level1*	52 - 70	initial nutrient content of each grid cell in the top layer (nutrient, energy and signal values are defined in abstract units)
Initial nutrient level2*	52 - 70	initial nutrient content of each grid cell in the deep/lower layer

Sigma3	30	diffusion parameter of the signal
Decay rate	0.003	decay rate of the signal
Cell (agent) parameters (defined separately for each strain)		
Initial energy	3	initial energy content of each cell
Nutrient uptake	1.6	defines how much nutrient is taken up by an active cell in a simulation step
Nutrient uptake efficiency	1	1: all of the nutrient taken up is converted to energy in the cell, values between 0 and 1
Division threshold	5	active cells divide after reaching this energy threshold
Division distance*	0.63 – 1.38	distance (in grid cells) between the daughter cell and its mother cell after division
Division distance decrease	inf	parameter for defining a time dependent decreasing (from 1 toward 0) multiplier of division distance. If the agar plate is drying with time, the colony growth slows down. It can be simulated via decreasing division distance. (This option was not used (was set to infinity) in the presented simulations.)
Metabolic energy	0.1	energy consumed by an active cell (from its internal energy pool) in each iteration
G0 threshold	0.3	active cells switch to stationary (G0) state if their energy levels fall below this threshold (this switch is reversible, if the energy level of a G0 cell exceeds this level, it switches back to active state)
G0 factor	0	multiplier for decreased nutrient uptake and metabolic energy on G0 state 0: G0 cells does not work (no energy uptake, metabolism) 1: no G0 effect
Death threshold	0.3	cells die if their energy levels fall below this threshold (their remaining energy returns to the top layer as nutrient)

Signal production	0	amount of signal produced by each active cell in every simulation step. Diverse signal effects can be specified, for example signal can control colony growth if it kills the cells or pushes them into G0 state above a certain threshold. (There is no signal production in the presented simulations.)
G0 signal production	0	multiplier (between 0 and 1) for signal production of G0 cells 0: no signal production in G0 1: no signal production decrease in G0
Signal effect1	80	the signal concentration affects the cells based on a sigmoid function. This parameter defines the change rate (slope) of the signal effect characteristics
Signal effect2	1	this parameter defines the signal value with 50% of the maximal signal effect (sets the range of effective signal concentrations)
Growth type	0	0: yeast-like growth 1: for filamentous growth: daughter cells stay close to the mother cell and cell lineages are forming a line of cells, via letting usually only the last cell to divide. (All simulations use yeast-like growth in the presented work.)
Division direction deviation	0	it defines the mean deviation of daughter cells from their mother cells in filamentous growth (0 for yeast-like growth), (it is needed for tortuous (but not random walk shaped) filaments, e.g.: 16 indicates $\pi/16$ radian mean deviation from the division direction of the mother cell)
Branching probability	0	probability of an inner cell in a filament for cell division (results in the emergence of branched filaments)
Initial cell number*	1 - 800	number of cells of a certain type (yeast strain) at the beginning of the simulation
Initial cell deviation*	10 - 25	radius (in grid cells) of the initial circular region (drop on the agar plate) populated with cells

Initial cell distribution	0.5	distribution of cells inside the initial drop (values e.g.: 0.5: uniform distribution; 0.25: more cells around the edge of the region; 1: more cell around the center of the region)
Initial drop center x	variable **	x coordinates (in grid cells) of the initial colony centers
Initial drop center y	variable **	y coordinates (in grid cells) of the initial colony centers

Supplementary Table 3: Parameter modifications between the simulations. Green: temporal parameters; yellow: spatial parameters of the environment or the agent; red: parameters defining material distribution in the environment; bold: fitted or fine-tuned values; italic: values calculated based on the setups of the wet lab experiments. See Supplementary Table 2 for the other parameters and for their descriptions.

	Simulation							
Parameter name	Fig.2 (1-6 colonies)	Fig.3b (24 colonies)	Fig.5 (unequal nutrient supply)	Fig.4a (dry condition)	Fig.4a (wet condition)	Fig.1c (single cell, small, bigger inoculation)	Fig.7a (dot near the colony)	Fig.6b (giant colony)
Initial nutrient level1	58.5	70	70 (350)	52	52	52	52	52
Initial nutrient level2	58.5	70	70 (350)	52	52	52	52	52
Sigma1	46.8	30	30	30	80	30	30	30
Sigma2	39	25	25	25	66.67	25	25	25
Division distance	0.795	0.63	0.63	0.63	1.38	0.63	0.63	0.63
Initial cell	23	10, 12, 17, 20, 23, 25	20	20	23	0, 12, 20	20	20

deviation								
Iteration number	480	228	480	312	312	336	864	2160
Plate size	620 x 620	870 x 540	870 x 540	620 x 620	620 x 620	620 x 620	620 x 620	900x900
Initial cell number	300	100, 200, 400, 800	300	300	300	1, 100, 300	300	300

Results: Understanding initial conditions of inoculation.

- The paper should state what the predictions are. Everything looks like a fitting to the data.

“We have tested our model to predict this unevenness for cases when statistics are not feasible or significant.” (lines 184-185)

We thank the referee for asking for this clarification. Indeed, in this section everything is fitted to the experiments. We have divided the Results section into two subsections now to emphasize that the first part is used for parameterization, after that the parameters were fixed. Predictions of the second subsection of Results were made with those fixed parameters.

We have removed the highlighted sentence as it created the confusion and discuss these in more details in the Discussion section. (lines 377-384)

- It is not clear if the parameters are fitted for each colony separately or all of them together. Clarification on the fitting process should be made.

“In order to mimic the conditions of regular high throughput OmiTray experiments when colonies grow close to each other for a shorter time, an agar layer was defined with high initial nutrient level, which is constantly consumed by cells at all spots. Diffusion values, cell division distances and initial nutrient amounts were used for fitting the colony sizes to the experimental data.” (lines 197-201)

Parameters were fitted on the average growth curve of all colonies with the same initial conditions (3-4 biological replicates). All growth curves were used during the fitting process (lines 218-219, 558-561).

- Figure 3 shows that experimental colonies are smooth, while simulations have a more anomalous contour. While the paper discusses the unevenness in the last

section of the results when it happens in the other direction, it does not mention anything in this one. This reviewer thinks that it may be a problem of the order of magnitude difference between simulations and experimental sizes. This should be clarified.

The referee is right, that this difference is probably the result of the differences in cell and agent numbers. Now we added a dedicated section in the discussion and also in the methods part to explain this difference and explain in the discussion that the model lack any simulation of extracellular matrix and agent numbers are lower than cell numbers, this could cause the observed difference in smoothness of colonies. We also point out that in the experimental case where the colony can grow really large (Fig 6) we also start to see the same effect on the experimental colony edges. (lines 396-421)

Results: How agar wetness affects colony size and structure?

- I consider this section very nice but finds it unfortunate that the fitting process is not clearly explained, and some important points are left unanswered.
 - How is the data in Figure 4c fitted? Are the Dry and Wet experiments fitted, and the other two predicted using this fitting? Are all four conditions fitted? In the latter case, why?
 - How do the fitted parameters adjust to the changing environments? The change in fitted parameters between conditions should be shown somewhere and commented upon.
 - The fitting only changes division distance and diffusion, however Figure 4b suggests that there is more proliferation of cells. Why has this not been fitted or predicted? If not included in a new fitting, it would be relevant to add some comments about it. This reviewer feels a bit awkward that this information appears in a Figure with all the other images being fittings, except from this one.

“Accordingly, ‘division distance’ and ‘diffusion’ parameters were optimized to describe these two phenomena. Simulation and experiment results both show that when colonies adjust to the changing environment, they form concentric rings of regions with different densities (Fig 4a), furthermore simulations capture colony growth dynamics in all conditions (Fig 4c).” (lines 256-263)

Two parameter sets were used in this experiment, one representing the dry, the other one representing the wet condition. They differ in the division distance of yeast cells and diffusion parameters of nutrients (higher diffusion coefficient and division distance in wet condition). The two parameter sets were switched in condition changes (wet to dry, dry to wet). The parameters were set to values that produce colony sizes comparable with the experimental results. So these results are not really predictions, but the dry and wet parameter sets were fixed between the same conditions in the four scenarios. This is now explained in the legend of Figure 4a ~~methods~~ section.

Agent number was added in panel c of figure 4 to clarify the scale difference, but show the similar trend.

Results: Predictions with the model

- The first sentence is very vague, and no further information is included. The paper should include a detailed description on what parameters have been used from all the above-mentioned fittings, how the simulated experiment has been set and the predictions clearly stated.

“Nutrients are rarely available in a standard, even concentration as commonly used in the lab. We have used the model and parameters optimized in the above-described experiments to predict colony growth in an environment with uneven nutrient distribution.” (lines 323-325)

We have rephrased this paragraph. Parameter values used are included in Supplementary Table 2 and 3 (shown above). (lines 274-276)

“Nutrients in nature are rarely available in a standard, even concentration as commonly used in the lab. We have used the model to predict colony growth in an environment with uneven nutrient distribution.”

- This introduction, again, is vague. It should be further detailed. If it is a prediction, they should state how they performed the simulations again.

“In our simulations, we realized that, when parameters of nutrient availability are perturbed, then the simulated colony could cover the whole simulation plane, independent of the size of the plane. This inspired us to test if we can grow a giant colony, which can cover the whole area of a Petri dish.” (lines 401-405)

We have rephrased this paragraph to make it more concise. (lines 323-329)

“In our parameter searching simulations, we noticed that, the simulated colony could cover the whole simulation plane. During the above presented experiments we have realized that increased agar wetness can maintain diffusion and provide nutrient availability for an extended period. This led to the hypothesis that periodic refeeding of the agar plate with liquid media can keep the colony growing unlimitedly (Fig 6.a).
“

Discussion:

- The following statement was not a prediction, according to the article, but a fitted property after experiments. Here you have changed the logical order or the results.

“Our model suggested and experiments confirmed...” (line 458)

The sentence was modified in the updated text.

- The paper does not state the efficiency of simulation capabilities of their software. Along with the comments on the code availability and documentation mentioned further below, I consider the following claims overstatements:

“We have created a powerful tool to simulate yeast colonies and used this model to explain the growth dynamics in numerous non-standard and non-constant environments.” (lines 486-487)

“Since the tool is easily extendable to multiple strains and interaction types, potential later applications can reveal further details of the mechanisms controlling yeast colony formation in mono- and mixed cultures.” (lines 489-492)

These sentences were modified in the text (phrases powerful and easily were deleted). We have also added supplementary table 4, where we show how simulation length scales with initial agent number. (lines 428-429, 432-434)

“We have created a software tool to simulate yeast colonies and used this model to explain the growth dynamics in numerous non-standard and non-constant environments.”

“Since the tool is extendable to multiple strains and interaction types, potential later applications can reveal further details of the mechanisms controlling yeast colony formation in mono- and mixed cultures.”

Minor concerns

- The paper goal does not seem to be an exploratory analysis of many conditions. The work already selects a set of factors to study, names seeding, humidity and nutrient intake (excluding many other relevant factors such as temperature, other nutrients' intake, competition...) so the statement that one of the goals of the article is to identify the crucial factors looks like an overstatement of the achievements of the article. When using the term “including”, it gives the impression that the article studied other factors that the work shown to be irrelevant to the colony growth; but the article just models and studies the mentioned factors.

*“**To identify the crucial factors** of colony growth dynamics, we have developed and parameterized a quantitative agent-based model of yeast colony growth by synergizing mathematical modeling with laboratory experiments.” (line 13-16)*

*“Through colony growth experiments and model fitting, we demonstrate the influence of various environmental factors, **including** moisture, nutrient availability, and initial colony inoculation conditions.” (line 17-20)*

We have rephrased these sentences. (lines 70-71, 143-144)

“...To be able to capture the dynamics and limiting factors of yeast colony growth we turned to agent-based modeling...”

“...The area over time was used to determine the following parameters of the model: cell division distances, nutrient diffusion rates, initial nutrient levels, and initial colony sizes. ...”

- It is not very clear what the authors meant with this sentence. Was it meant to be a “However”, maybe?

“Growing yeast colonies and using colony size as an indicator of fitness is a daily routine in many research labs. Although, many details on the mechanism and dynamics of colony growth are less understood.” (lines 37-39)

The referee is correct. We have rephrased the mentioned paragraph. (lines 37-40)

“...These days, growing yeast colonies and using colony size as an indicator of fitness is a daily routine in many research laboratories. Still, the underlying cellular processes which affect colony growth are far less understood. ...”

- It may have been nice to quantify the shape parameter, or at least comment how it could have been used quantitatively.

“One can observe that colony shapes change from circular to a less and less circular shape due to the proximity of neighboring colonies and the edge of the plate. This feature was also present in our simulations (Fig. 2b).” (Lines 153-157)

We have looked at possible ways of quantifying the shape parameter. While some trends can be observed, these did not prove to be robust for publication. Standard measures like the ‘Round’ or ‘Circularity’ parameters built into FIJI have failed to represent the changes perceived visually for two main reasons. The first, and more challenging is the fact that the growth happens asymmetrically. This results in a more circular overall shape (at the 6 colonies per plate than the 3 colonies one), however this shifts the center of mass away from the original location of inoculation. The other challenge is the more general one, considering the difficulties in measuring the segmented perimeter correctly. In our case a small sparkle on the colony side can alter the measured perimeter significantly.

The ideal measure to quantify the shape parameter should take into consideration the inoculation location and consider differences in radial growth into the different directions. While it certainly is possible to create such a measure, it might be arbitrary.

- Although this reviewer has read the materials section and seen the cell counting process; it is still unclear to them how you control cell number at the seeding time. A further clarification would be nice.

“Standard rectangular plate format (OmiTray) was used to investigate two initial inoculation parameters: droplet size and cell number.” (lines 186-187)

We have supplemented the relevant Results and Methods sections: (lines 190-193, 462-466)

“Inoculation droplet size was varied by changing the volume of the drops while cell numbers in these drops were fixed by changing the concentrations accordingly. The other variable, the initial cell count, was varied by changing the concentration, while keeping the drop sizes.”

“Cell count was determined from the overnight preculture using a hemocytometer and appropriate dilutions were prepared for each concentration according to the combination of droplet size (0.5, 1, 2, 3, 4, 5 μ l) and initial cell count (1000, 2000, 4000, and 8000) pairs. I.e. preculture was diluted to 2×10^6 cells per ml for the 0.5 μ l droplet for 1000 initial cell numbers, this same dilution was used for the 1 μ l-2000 cells spot as well. Droplets were spotted using a grid template placed under the plate. The layout was shuffled for control plates.”

- Figure 3 is somehow confusing, as the a and b subsections show initial droplet x initial cell count, but subcaption c shows time x area. Maybe putting the a and b axis labels in the top and right sides of the figure may help the reader to see the equivalence.

We thank for the recommendation. We have edited the layout of Figure 3 for easier understanding.

- There is no evidence in the distribution of active cells since, as mentioned in the same paper, there are no observables of this distribution. The authors should reconsider if this is a prediction of the model, as the authors state. From the sentence, I interpret it as a known true fact.

“On all plates, colony growth was observable towards ‘free’ areas, from which directions further nutrients became available by diffusion. This supports the power of simulation in uneven environments, but simulation can give an additional insight that is difficult to measure experimentally, which is the distribution of active cells. By plotting actively dividing cells, we can predict where growth is expected to happen (Fig. 5c).” (lines 340-345)

Even though there is no direct evidence of actively dividing cells, the range expansion from 13 to 20 days can only happen where there are actively dividing cells. We considered this range expansion as indirect evidence. Figure 5c was modified to show active cells at day 13. This way we consider the model’s actively dividing cell distribution at 13 days as a prediction for the following range expansion.

Data availability.

I have not been able to find the data of the article, nor is it mentioned anywhere if it will be published anywhere at some point

Data availability statement has been added. Data to the final arrangement of figures are available in the same repository as the codes. (lines 568-570)

“All data supporting the figures are stored together with their codes on the Github repository https://github.com/CsikaszNagyLab/yeast_colony_growth_model “

Code availability.

The repository is public and well organized. However, the documentation is very limited and only contains a minimal set of instructions to run the simulations performed in the article. The code for the data analysis is not provided in the repository. I am concerned about the claims made by the paper that this software could be used to perform custom analysis and extension, as no information on how to use the code outside the provided simulations is given. I am also concerned about the programming language being used. While the code of the article is publicly available, MATLAB is a proprietary software that is not common in many laboratories nowadays, and this limit the accessibility of the results to a large audience. Analyzing the code script, I see no technical burdens to implement this code in an open language such as Python, C or Julia. This reviewer is aware of the complexities of programming and is not asking for migrating the code to a different language for publication; but wants to remark, for future work and for the sake of code transparency in science, that the authors reconsider the programming language that they choose to use.

The readme file was extended with some additional instructions about the basic use of the model. One can use the current model with setups similar to the ones described in the article. In addition the model is capable of handling different yeast strains in its current state by defining strains with different parameters in the input file.

Reviewer #2 (Remarks to the Author):

In this paper, the authors presented a modeling approach to analyze different environmental factors on yeast colony growth. The growth of yeast cells in a defined Petri dish environment is a spatio-temporally complex. The paper has a simple algorithm that connects several factors including nutrient availability and gradient, and inoculation size to colony morphology, and offer good insights on how these factors shape the morphology all together.

My comments and suggestions for revisions are as below:

(1) Fig 1a: The manuscript currently contains no mathematical equations, nor detailed explanations on how parameters are fitted and trained. It is not clear how different parameters are connected, and why the choice of unitless parameters, and how certain threshold parameters such as cell death and division were chosen rationally (whether it's based on biological realism). The authors also mentioned about the possibility of adding stochastic cell birth-death processes. It's worthwhile to explain how. Given that modeling is the major innovation in this paper, I would suggest a major revision on this part.

Equations about the nutrient flow (lines 512-522) and parameter fit (lines 558-565) objective function were added.

The list of fitted parameters and some explanation about their effects and connections was added to the end of the "Model parametrization" section (lines 145-172). In addition Supplementary Table 2 was updated with default parameter values, ranges and notes on their connection. Supplementary Table 3 was added with experiment specific parameter values. Some parameters, mainly the cell behavior and cell metabolism related ones, are unitless because they were not measured experimentally, and they do not directly represent *in vivo* biological processes, since the agents of the computational model do not represent single yeast cells rather a collection of nearby cells. Their values are not based on biological reality, they were set in a way to achieve realistic colony sizes and shapes. Some time and space related parameters are defined in simulation grid units or in timesteps. Their values can be converted into real life quantities. 1 simulation step is around 1 hour and one grid cell is around 55.6 mm².

Stochastic changes could be incorporated to the model by defining a probability of events (e.g. cell division, death, G0 transition, nutrient uptake) when their conditions are satisfied. This possibility improvement is noted at the revised discussion section (lines 417-419).

Supplementary table 2: Input parameters and output descriptions of the model.
Green: temporal parameters; yellow: spatial parameters of the environment or the agent; red: parameters defining material distribution in the environment; blue: cell metabolism and energetics related parameters; bold: fitted or fine-tuned parameters.

Input parameters are defined in a .csv file. The simulations produce outputs for on-the-fly data visualizations, but also save .mat and .tsv data files for post processing.

Parameter name	Used value(s)	Description
General and visualization parameters		
Iteration number*	228 - 2160	length of the simulation (in simulation steps) 1 step~1 hour with the default settings
Time step	1	time resolution
Draw	0,1	1: visualizations during the simulation are on 0: no visualizations during the simulation
Visualization steps	20	frequency of updating the graphical output during the simulation (x indicates update in every xth steps)
Environmental (plate, agar and nutrient related) parameters		
Plate x*	620 - 900	x dimension of the plate (in grid cells)
Plate y*	620 - 900	y dimension of the plate (in grid cells) 1 grid cell ~ 55.6 mm ² with default settings
Boundary condition	1	0: periodic boundary condition 1: fixed boundary condition (used in the presented work)
Sigma1*	30 – 80	diffusion parameter of the top nutrient layer
Sigma2*	25 - 66.67	diffusion parameter of the deep/lower nutrient layer
Flow rate	0.01	nutrient exchange rate between the two nutrient layers
Initial nutrient level1*	52 - 70	initial nutrient content of each grid cell in the top layer (nutrient, energy and signal values are defined in abstract units)
Initial nutrient level2*	52 - 70	initial nutrient content of each grid cell in the deep/lower layer
Sigma3	30	diffusion parameter of the signal
Decay rate	0.003	decay rate of the signal

Cell (agent) parameters (defined separately for each strain)		
Initial energy	3	initial energy content of each cell
Nutrient uptake	1.6	defines how much nutrient is taken up by an active cell in a simulation step
Nutrient uptake efficiency	1	1: all of the nutrient taken up is converted to energy in the cell, values between 0 and 1
Division threshold	5	active cells divide after reaching this energy threshold
Division distance*	0.63 – 1.38	distance (in grid cells) between the daughter cell and its mother cell after division
Division distance decrease	inf	parameter for defining a time dependent decreasing (from 1 toward 0) multiplier of division distance. If the agar plate is drying with time, the colony growth slows down. It can be simulated via decreasing division distance. (This option was not used (was set to infinity) in the presented simulations.)
Metabolic energy	0.1	energy consumed by an active cell (from its internal energy pool) in each iteration
G0 threshold	0.3	active cells switch to stationary (G0) state if their energy levels fall below this threshold (this switch is reversible, if the energy level of a G0 cell exceeds this level, it switches back to active state)
G0 factor	0	multiplier for decreased nutrient uptake and metabolic energy on G0 state 0: G0 cells does not work (no energy uptake, metabolism) 1: no G0 effect
Death threshold	0.3	cells die if their energy levels fall below this threshold (their remaining energy returns to the top layer as nutrient)
Signal production	0	amount of signal produced by each active cell in every simulation step. Diverse signal effects can be specified, for example signal can control colony growth if it kills the cells or pushes them into G0

		state above a certain threshold. (There is no signal production in the presented simulations.)
G0 signal production	0	multiplier (between 0 and 1) for signal production of G0 cells 0: no signal production in G0 1: no signal production decrease in G0
Signal effect1	80	the signal concentration affects the cells based on a sigmoid function. This parameter defines the change rate (slope) of the signal effect characteristics
Signal effect2	1	this parameter defines the signal value with 50% of the maximal signal effect (sets the range of effective signal concentrations)
Growth type	0	0: yeast-like growth 1: for filamentous growth: daughter cells stay close to the mother cell and cell lineages are forming a line of cells, via letting usually only the last cell to divide. (All simulations use yeast-like growth in the presented work.)
Division direction deviation	0	it defines the mean deviation of daughter cells from their mother cells in filamentous growth (0 for yeast-like growth), (it is needed for tortuous (but not random walk shaped) filaments, e.g.: 16 indicates $\pi/16$ radian mean deviation from the division direction of the mother cell)
Branching probability	0	probability of an inner cell in a filament for cell division (results in the emergence of branched filaments)
Initial cell number*	1 - 800	number of cells of a certain type (yeast strain) at the beginning of the simulation
Initial cell deviation*	10 - 25	radius (in grid cells) of the initial circular region (drop on the agar plate) populated with cells
Initial cell distribution	0.5	distribution of cells inside the initial drop (values e.g.: 0.5: uniform distribution; 0.25: more cells)

		around the edge of the region; 1: more cell around the center of the region)
Initial drop center x	variable **	x coordinates (in grid cells) of the initial colony centers
Initial drop center y	variable **	y coordinates (in grid cells) of the initial colony centers

Supplementary Table 3: Parameter modifications between the simulations. Green: temporal parameters; yellow: spatial parameters of the environment or the agent; red: parameters defining material distribution in the environment; bold: fitted or fine-tuned values; italic: values calculated based on the setups of the wet lab experiments. See Supplementary Table 2 for the other parameters and for their descriptions.

Parameter name	Simulation							
	Fig.2 (1-6 colonies)	Fig.3b (24 colonies)	Fig.5 (unequal nutrient supply)	Fig.4a (dry condition)	Fig.4a (wet condition)	Fig.1c (single cell, small, bigger inoculation)	Fig.7a (dot near the colony)	Fig.6b (giant colony)
Initial nutrient level1	58.5	70	70 (350)	52	52	52	52	52
Initial nutrient level2	58.5	70	70 (350)	52	52	52	52	52
Sigma1	46.8	30	30	30	80	30	30	30
Sigma2	39	25	25	25	66.67	25	25	25
Division distance	0.795	0.63	0.63	0.63	1.38	0.63	0.63	0.63
Initial cell deviation	23	10, 12, 17, 20, 23, 25	20	20	23	0, 12, 20	20	20

Iteration number	480	228	480	312	312	336	864	2160
Plate size	620 x 620	870 x 540	870 x 540	620 x 620	620 x 620	620 x 620	620 x 620	900x900
Initial cell number	300	100, 200, 400, 800	300	300	300	1, 100, 300	300	300

(2) Fig 1b: Based on the picture it doesn't seem like the model simulation recreated the volcano crater size. Suggest to plot the colony in cross-sectional view.

We thank the referee for this suggestion. We have added intensity plots of the cross-sectional view in Figure 1 b and c (Experiment and simulation). For better illustration we have changed the 2 days old pictures for 8 days old pictures where the volcano shape is more evident. These represent intensities of the colonies. While there are limitations in this view regarding the height, the crater size can indeed be observed better.

(3) Fig 3: Suggest to provide a quantitative measurement of the effects of initial cell count and droplet size. It's not clear the degree of interactions between the two parameters on the colony area.

Supplementary figure 3 was added to illustrate that despite other noise factors, initial droplet size does determine colony size while initial cell count does not.

We have noticed that drops with 8000 cells were further away from the edge of the plate, thus they had a larger area to collect nutrients from. This result highlights the importance of modeling, where after the experiments an error was discovered, which misled the data analysis. In this case statistics failed due to the systematic experimental error, while modeling could replace the repetition of the whole experiment and provide a proper explanation.

Supplementary Figure 3 Impact of the inoculation condition on the colony size at 14 days Area is shown by initial cell number and droplet size in both orders. While increasing trend can be observed for Initial droplet size, it is nearly flat for initial cell count. Colonies started from 8000 cells (most of which were on the edge of the plate) stand out from this trend.

(4) It's confusing how agent numbers correlate with the cell numbers, and why there's a magnitude of difference between the number of cells/agents in simulation and in experiments. Why is the model not being able to fit this? If the author is predicting the effect of initial cell number to the colony area in Fig 3, the cell number parameter seems to be very important in the model.

Colony areas are easier to measure experimentally in a reliable way than cell numbers. The three-dimensional structure of the colonies and the metabolic state of the cells (actively dividing, G0, or dead) could make these estimations even less reliable. This is the reason why we have decided to characterize colonies based on their sizes rather than their cell counts and used size related parameters for fitting. Theoretically, cell counts could be also fitted to the experiments (knowing the metabolic parameters of the cells), but handling that many agents is not computationally efficient (less agents can reproduce the studied metrics: colony shapes and sizes). We explain this now in the methods section (lines 535-540) and show the statistics on colony size predictions with various initial agent numbers in simulations on table S4.

Supplementary Table 4: Effect of agent numbers to the final colonies. Statistics of 14 day long simulations started from different (30, 300 (default), 3000) initial agent sizes. Agent metabolism parameters were scaled up 10 times in the 30 agent scenario and scaled down 10 times in the 3000 agent scenario. The results demonstrate that the 30 initial agent case results in smaller and more irregular colony size, but the 3000 initial agent case produce similar colony size and shape (slightly smoother) to the

default 300 initial agent scenario, but with longer simulation time and higher memory requirement.

Initial agent number	30	300	3000
Final agent number	45513	596792	6216080
Final agent number/initial agent number	1517.1	1989.31	2072.03
Final colony area (grid point)	6434	16686	19346
Simulation time (min)	0.4	5.7	69.3
Size of stored data output from a simulation (MB)	10.6	123	1220
Final colony figure			
(5) Fig 4: top panel should be removed. It is redundant.

Thank you for noticing. We have removed it.

(6) Fig 5: The arrangement of colonies seem arbitrary. Explain.

Layout was designed to leave out first, last and in between colonies as well (measured from nutrient supply side). This way the symmetry and expected symmetrical results could have been broken and an intact column of 4 colonies was also included. We have complemented the 'Results' section with the following description (lines 283-286):

“Inoculation pattern was designed to leave out a colony from each row while keeping 2 full columns. This enables us to test how a growing colony in upper gradient might influence growth behind at a lower nutrient region. This is also another validation test for the model.”

(7) Fig 6: How was the media replenished? Does the formation of “rings” correlate with the media renewal? In Fig 6c, the authors added a drop on and next to an aging colony, but didn't explain the rationale of this experiment nor the interpretation of it.

Media was replenished through 4 smaller slots that were cut on the agar before inoculation. Liquid YPD was added through these slots biweekly. Supplementary Figure 4 has been added to explain the experimental procedure (lines 724-726). The ring formation does

correlate with the addition of fresh media. Both nutrient shortage and drying of the agar has influenced this. The biweekly refeed still left some room for drying as time passed.

In Fig 6c (moved to Figure 7 in the revised version), the 2 droplets were added for the purpose of qualitative representation only, hence a distant but non symmetric arrangement was chosen. Now we have also expanded on the interpretation of these results (lines 347-355).

Reviewers' comments:

Reviewer #1 (Remarks to the Author):

The authors have performed a length revision of the article, going over the major concerns on the paper that restricted its publication that solved my doubts.

I find it now suitable for publication now.

Gabriel Torregrosa Cortés

Reviewer #2 (Remarks to the Author):

In the revised article, the authors addressed key concerns around inadequate literature review in the introduction section, and lack of details in model parameterization and fitting.

The revised introduction gives a better context on the importance of the study. The authors surveyed prior studies on colony morphology and other agent-based models in studying bacterial colony morphology. However, it's worth noting that phenomenological models have already been developed extensively to study cell morphologies of various cell types in different environmental conditions, which include several reaction-diffusion models [1,2] and individual-based models [3,4]. In particular, models based on nutrient limitation and diffusion have been shown in many examples to reproduce experimentally observed spatial patterns [5-8]. Therefore, the scientific novelty of applying the proposed agent-based model and how it is better than (or differs from) other phenomenological models are not very clear. It is also not evident how such phenomenological models can be useful to provide a mechanistic understanding of cell formation (e.g. metabolism, genetics, microbial interactions) as claimed by the authors (line 433).

Regarding the revision on the model details and parameterization, the authors made a good effort on providing the parameter values, ranges and their connections. It is still unclear how the initial values of these parameters were selected, and whether changing the initial selection could impact the parameter fitting. For estimating the six parameters, since the experimental measurement data was sparse (line 562) and the parameter combinatorial search space is rather large, I would like to understand how was parameter scan done and whether issues such as overfitting could occur. The authors have added the fitting objective (line 558) but the fitting algorithm remains unexplained. Since the parameters here are not based on biological realisms according to the authors, it is important to thoroughly study the parameter choice and its impact on the downstream application of this model.

Reference:

- 1 Lega, J., & Passot, T. (2003). Hydrodynamics of bacterial colonies: a model. *Physical Review E*, 67(3), 031906.
- 2 Golding, I., Kozlovsky, Y., Cohen, I., & Ben-Jacob, E. (1998). Studies of bacterial branching growth using reaction-diffusion models for colonial development. *Physica A: Statistical Mechanics and its Applications*, 260(3-4), 510-554.
- 3 Vassallo, L., Hansmann, D., & Braunstein, L. A. (2019). On the growth of non-motile bacteria colonies: an agent-based model for pattern formation. *The European Physical Journal B*, 92, 1-8.
- 4 Golden, A., Dukovski, I., Segrè, D., & Korolev, K. S. (2022). Growth instabilities shape morphology and genetic diversity of microbial colonies. *Physical Biology*, 19(5), 056005.
5. Wakita, J. I., Komatsu, K., Nakahara, A., Matsuyama, T., & Matsushita, M. (1994). Experimental investigation on the validity of population dynamics approach to bacterial colony formation. *Journal of the Physical Society of Japan*, 63(3), 1205-1211.

6. Tokita, R., Katoh, T., Maeda, Y., Wakita, J. I., Sano, M., Matsuyama, T., & Matsushita, M. (2009). Pattern formation of bacterial colonies by escherichia coli. *Journal of the Physical Society of Japan*, 78(7), 074005-074005.
7. Matsushita, M., & Fujikawa, H. (1990). Diffusion-limited growth in bacterial colony formation. *Physica A: Statistical Mechanics and its Applications*, 168(1), 498-506.
8. Müller, J., & Van Saarloos, W. (2002). Morphological instability and dynamics of fronts in bacterial growth models with nonlinear diffusion. *Physical Review E*, 65(6), 061111.

Reviewers' comments:

Reviewer #1 (Remarks to the Author):

The authors have performed a length revision of the article, going over the major concerns on the paper that restricted its publication that solved my doubts.

I find it now suitable for publication now.

Gabriel Torregrosa Cortés

Dear Gabriel,

We thank you for the thorough review, which helped us to improve our manuscript. We are glad we managed to resolve your doubts.

Reviewer #2 (Remarks to the Author):

In the revised article, the authors addressed key concerns around inadequate literature review in the introduction section, and lack of details in model parameterization and fitting.

The revised introduction gives a better context on the importance of the study. The authors surveyed prior studies on colony morphology and other agent-based models in studying bacterial colony morphology. However, it's worth noting that phenomenological models have already been developed extensively to study cell morphologies of various cell types in different environmental conditions, which include several reaction-diffusion models [1,2] and individual-based models [3,4]. In particular, models based on nutrient limitation and diffusion have been shown in many examples to reproduce experimentally observed spatial patterns [5-8]. Therefore, the scientific novelty of applying the proposed agent-based model and how it is better than (or differs from) other phenomenological models are not very clear. It is also not evident how such phenomenological models can be useful to provide a mechanistic understanding of cell formation (e.g. metabolism, genetics, microbial interactions) as claimed by the authors (line 433).

Thank you for directing our attention to the referenced papers, most of these are now discussed in the manuscript either in the introduction to position our work on in the discussion to explain how our model can be further developed by incorporating more details on the physics of microbial colony formation.

Introduction: (lines 51-54)

“Mathematical models have helped the interpretation of many of the above listed observations on microbial colony growth^{13,19} and others also focused on integrating knowledge to capture colony growth dynamics²⁰⁻²² and colony structure emergence²³⁻²⁶, even incorporating physical forces²⁷⁻²⁹.”

Here we have also added a sentence to explain the differences between models of bacterial and yeast colony formation: (lines 58-62)

“While there is a high level of similarity between the structure of bacterial and yeast colonies, there are some considerable differences in how these are formed. Motility of bacterial cells is missing in yeasts, and their cell division characteristics and periods are also dissimilar. These differences need to be taken into consideration before applying the above bacterial models for yeast colony formation.”

Discussion: (lines 406-410)

“There are examples for models with considerations of physical forces dominantly in the bacterial context^{27,49,50} and few more specific for yeast^{51,52}. Implementing these additional constraints to colony development could make our model even more realistic, however it would increase its computational requirements (or reduce the size of colony that could be simulated) and the required experimental data at the same time.”

Our model was developed to reproduce, and in some cases predict, the size and area of yeast colonies in nutrient rich (even if uneven) environments as well as the impact of colonies growing nearby each other. This was done in a combination of experiments and models focusing on specific questions about colony growth. We cannot really compare all earlier developed models with our experimental data as most are focused on a single colony and consider bacterial movements of cells. Our purpose built, agent-based model is easily extendable with metabolic networks within cells or production of any diffusing material that could facilitate communication and interactions between cells/agents. Thus we believe the model as well as experimental results will influence further research in the yeast systems biology field and give a generic view on microbial growth dynamics.

Regarding the revision on the model details and parameterization, the authors made a good effort on providing the parameter values, ranges and their connections. It is still unclear how the initial values of these parameters were selected, and whether changing the initial selection could impact the parameter fitting. For estimating the six parameters, since the experimental measurement data was sparse (line 562) and the parameter combinatorial search space is rather large, I would like to understand how was parameter scan done and whether issues such as overfitting could occur. The authors have added the fitting objective (line 558) but the fitting algorithm remains unexplained. Since the parameters here are not based on biological realisms according to the authors, it is important to thoroughly study the parameter choice and its impact on the downstream application of this model.

The initial parameter values were set to produce realistic colony sizes and shapes in reasonable simulation time (with a reasonable number of agents). Different initial parameter sets would result in different fitted parameter values.

The fitting algorithm was the MATLAB implementation of the Nelder-Mead simplex search algorithm (lines 561-564). This part was extended in the Methods section:

“The fitting of model parameters to experimental data was performed in MATLAB using a built-in Nelder-Mead simplex search algorithm (fminsearch) 52 initialized from a set of intuitively defined initial conditions producing realistic colony sizes and shapes with a reasonable number of agents (and thereby reasonable simulation times).“

The experimental data was sparse in the temporal domain, but because of the nature of colony growth (random fluctuations, fast changes in colony sizes are unlikely, rather indicate imaging problems) these limited numbers of images could capture the developmental dynamics of the yeast strains. Overfitting this sparse data was avoided by fixing as many parameter values as we could while fitting all observations. The remaining 6 fitted parameters are affecting the measured variables differently and further reduction of tested parameter space was not possible. We used the mean values of the parallel experimental data points which also reduces the risk of overfitting.

Reference:

- 1 Lega, J., & Passot, T. (2003). Hydrodynamics of bacterial colonies: a model. *Physical Review E*, 67(3), 031906.
- 2 Golding, I., Kozlovsky, Y., Cohen, I., & Ben-Jacob, E. (1998). Studies of bacterial branching growth using reaction–diffusion models for colonial development. *Physica A: Statistical Mechanics and its Applications*, 260(3-4), 510-554.
- 3 Vassallo, L., Hansmann, D., & Braunstein, L. A. (2019). On the growth of non-motile bacteria colonies: an agent-based model for pattern formation. *The European Physical Journal B*, 92, 1-8.
- 4 Golden, A., Dukovski, I., Segrè, D., & Korolev, K. S. (2022). Growth instabilities shape morphology and genetic diversity of microbial colonies. *Physical Biology*, 19(5), 056005.
5. Wakita, J. I., Komatsu, K., Nakahara, A., Matsuyama, T., & Matsushita, M. (1994). Experimental investigation on the validity of population dynamics approach to bacterial colony formation. *Journal of the Physical Society of Japan*, 63(3), 1205-1211.
6. Tokita, R., Kato, T., Maeda, Y., Wakita, J. I., Sano, M., Matsuyama, T., & Matsushita, M. (2009). Pattern formation of bacterial colonies by escherichia coli. *Journal of the Physical Society of Japan*, 78(7), 074005-074005.
7. Matsushita, M., & Fujikawa, H. (1990). Diffusion-limited growth in bacterial colony formation. *Physica A: Statistical Mechanics and its Applications*, 168(1), 498-506.
8. Müller, J., & Van Saarloos, W. (2002). Morphological instability and dynamics of fronts in bacterial growth models with nonlinear diffusion. *Physical Review E*, 65(6), 061111.